# The Choice of Normalization Influences Shrinkage in Regularized Regression

**Johan Larsson**  *jola@math.ku.dk*
*Department of Mathematical Sciences, University of Copenhagen*
*Department of Statistics, Lund University*

**Jonas Wallin**  *jonas.wallin@stat.lu.se*
*Department of Statistics, Lund University*

**Reviewed on OpenReview:** *https://openreview.net/forum?id=6xKyDBIwQ5*

## Abstract

Regularized models are often sensitive to the scales of the features in the data and it has therefore become standard practice to normalize (center and scale) the features before fitting the model. But there are many different ways to normalize the features and the choice may have dramatic effects on the resulting model. In spite of this, there has so far been no research on this topic. In this paper, we begin to bridge this knowledge gap by studying normalization in the context of lasso, ridge, and elastic net regression. We focus on binary features and show that their class balances (proportions of ones) directly influences the regression coefficients and that this effect depends on the combination of normalization and regularization methods used. We demonstrate that this effect can be mitigated by scaling binary features with their variance in the case of the lasso and standard deviation in the case of ridge regression, but that this comes at the cost of increased variance of the coefficient estimates. For the elastic net, we show that scaling the penalty weights, rather than the features, can achieve the same effect. Finally, we also tackle mixes of binary and normal features as well as interactions and provide some initial results on how to normalize features in these cases.

## 1 Introduction

When modeling high-dimensional data where the number of features ($p$) exceeds the number of observations ($n$), it is impossible to apply classical statistical models such as standard linear regression since the design matrix $\boldsymbol{X}$ is no longer of full rank. A common remedy to this problem is to *regularize* the model by adding a penalty term to the objective that punishes models with large coefficients. The resulting problem takes the following form:

$$\underset{\beta_0\in\mathbb{R},\boldsymbol{\beta}\in\mathbb{R}^p}{\text{minimize}}\ g(\beta_0, \boldsymbol{\beta}; \boldsymbol{X}, \boldsymbol{y}) + h(\boldsymbol{\beta}), \tag{1}$$

where $\boldsymbol{y}$ is the response vector, $\boldsymbol{X}$ the design matrix, $\beta_0$ the intercept, and $\boldsymbol{\beta}$ the coefficients. Furthermore, $g$ is a data-fitting function that attempts to optimize the fit to the data and $h$ is a penalty that depends only on $\boldsymbol{\beta}$. Two common penalties are the $\ell_1$ norm and squared $\ell_2$ norm penalties, which if $g$ is the standard ordinary least-squares objective, represent the lasso (Tibshirani, 1996; Santosa & Symes, 1986; Donoho & Johnstone, 1994) and ridge (Tikhonov) regression respectively.

These penalties depend on the magnitudes of the coefficients, which means that they are sensitive to the scales of the features in $\boldsymbol{X}$. To avoid this, it is common to *normalize* the features before fitting the model by shifting and scaling each feature by some measures of their locations and scales, respectively. For some problems it is possible to arrive at these measures by contextual knowledge of the data at hand. In most cases, however, they must be estimated. A popular strategy is to use the mean and standard deviation of each feature as location and scale factors respectively, which is called *standardization.*

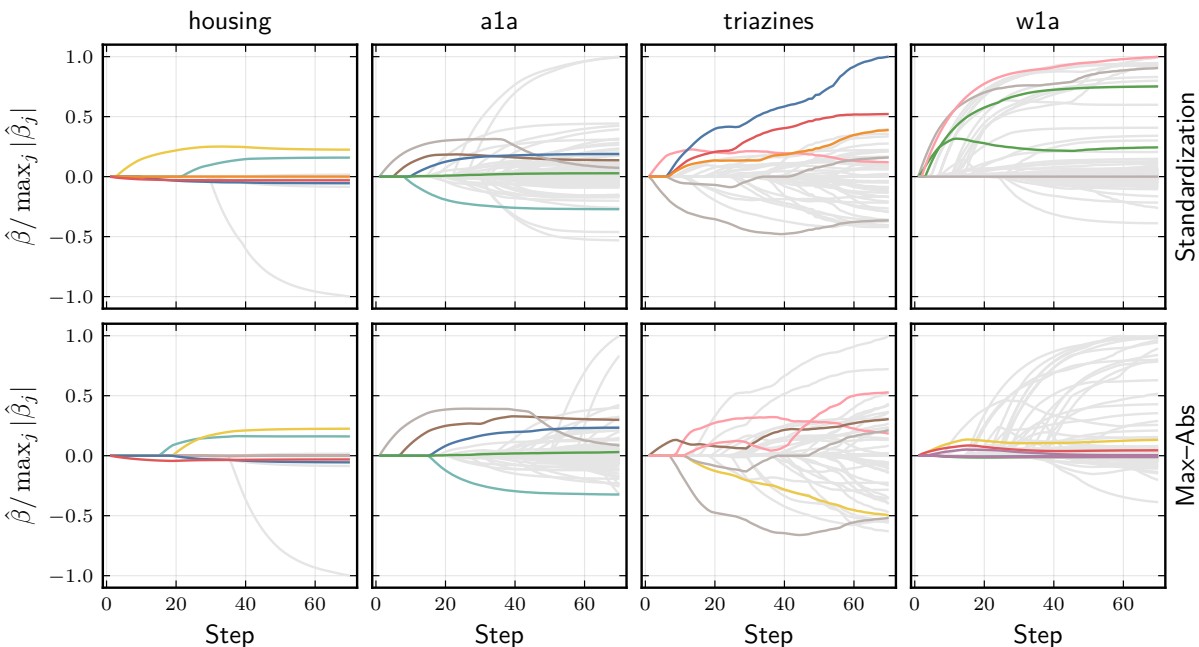

Figure 1: Lasso paths for real datasets using two types of normalization: standardization and maximum absolute value normalization (max–abs). For each dataset, we have colored the coefficients if they were among the first five to become non-zero under either of the two normalization schemes. The $x$-axis shows the steps along the regularization path and the $y$-axis the estimated coefficients normalized by the maximum magnitude of the coefficients in each case. See Section G for more information about datasets used here.

The choice of normalization may, however, have consequences for the estimated model. As a first example of this, consider Figure 1, which displays the regularization paths for the lasso[1] for four datasets: housing (Harrison & Rubinfeld, 1978), a1a (Becker & Kohavi, 1996; Platt, 1998), triazines (King et al., 1995; Hirst et al., 1994), and w1a (Platt, 1998), and two types of normalization.

In the figure, we have colored the lines corresponding to features that were among the first five to enter the model in either of the two type of normalization schemes. Note that the choice of normalization result in different sets of features being selected as well as different coefficient estimates. This is especially striking in the case of triazines and w1a.

To illustrate this further, we show the estimated coefficients for the same datasets after having fitted the lasso with a penalty strength ($\lambda$) set by 5-fold cross-validation repeated 5 times on a 50% training data subset. The estimated coefficients for the 50% held-out test data are shown in Table 1. We see that the estimated coefficients on the test set are different between the two cases, especially for the triazines datasets, where the two normalization schemes disagree completely and the max–abs normalization results in all coefficients being zero.

In spite of this apparent connection between normalization and regularization, there has so far been almost no research on the topic. And in its absence, the choice of normalization is typically motivated by computational concerns or by being "standard". This is problematic since the effects of normalization are unknown and because there exists no natural choice for many types of data. In particular, there is no obvious choice for binary features (where each observation takes either of two values). In this paper we begin to bridge this knowledge gap by studying normalization in the context of three particular cases of the regularized problem in Equation (1): the lasso, ridge, and elastic net (Zou & Hastie, 2005). The latter of these, the elastic net, is

---

[1]The estimated coefficients of the lasso as the penalty strength is varied from a large-enough value for all coefficients to be zero to a low value at which the model is almost saturated.

Table 1: Estimated lasso coefficients on test sets, with $\lambda$ set from 5-fold cross-validation repeated 5 times. $\hat{\boldsymbol{\beta}}_{\text{std}}$ and $\hat{\boldsymbol{\beta}}_{\text{max–abs}}$ show the coefficients based on normalizing the design matrix with standardization and maximum absolute value (max–abs) normalization respectively. We show the five largest coefficients (in magnitude) for the standardization case and the respective coefficients for the max–abs case. In each case, we present the coefficients on the original scale of the features. See Section G for more information about these datasets.

| housing | | a1a | | triazines | | w1a | |
|---|---|---|---|---|---|---|---|
| $\hat{\beta}_{\text{std}}$ | $\hat{\beta}_{\text{max–abs}}$ | $\hat{\beta}_{\text{std}}$ | $\hat{\beta}_{\text{max–abs}}$ | $\hat{\beta}_{\text{std}}$ | $\hat{\beta}_{\text{max–abs}}$ | $\hat{\beta}_{\text{std}}$ | $\hat{\beta}_{\text{max–abs}}$ |
| $-0.63$ | $-0.68$ | $0.54$ | $0.54$ | $0.17$ | $0.0$ | $1.8$ | $0.0$ |
| $-1.4$ | $-0.78$ | $0.33$ | $0.52$ | $0.069$ | $0.0$ | $1.8$ | $0.78$ |
| $0.27$ | $0.0$ | $-0.39$ | $-0.51$ | $0.028$ | $0.0$ | $1.8$ | $0.63$ |
| $-0.99$ | $-0.34$ | $0.31$ | $0.32$ | $0.071$ | $0.0$ | $1.4$ | $0.080$ |
| $2.8$ | $3.1$ | $0.18$ | $0.23$ | $0.029$ | $0.0$ | $1.7$ | $0.0$ |

a generalization of the previous two, and is represented by the following optimization problem:

$$\underset{\beta_0\in\mathbb{R},\boldsymbol{\beta}\in\mathbb{R}^p}{\text{minimize}}\ \frac{1}{2}\|\boldsymbol{y} - \beta_0 - \tilde{\boldsymbol{X}}\boldsymbol{\beta}\|_2^2 + \lambda_1\|\boldsymbol{\beta}\|_1 + \frac{\lambda_2}{2}\|\boldsymbol{\beta}\|_2^2, \tag{2}$$

where setting $\lambda_1 = 0$ results in ridge regression and setting $\lambda_2 = 0$ results in the lasso. Our focus in this paper is on binary data and we pay particular attention to the case when they are imbalanced, that is, have relatively many ones or zeroes. In this scenario, we demonstrate that the choice of normalization directly influences the estimated coefficients and that this effect depends on the particular combination of normalization and regularization. For a first illustration of this, see Figure 2, which corresponds to a small experiment in which we have generated two binary features with class balances 0.5 and 0.9, respectively, and fit full lasso and ridge regression paths. The coefficients when generating the data were set to one in both cases. The type of normalization affects the sizes of the resulting coefficients. This, for instance, leads to opposite conclusions being drawn about feature importance depending on whether L1 or Max–Abs normalization is used. For instance, at the point where the coefficient of the balanced feature is 0.5, the coefficient of the imbalanced features is roughly 0.75 with L1 normalization compared to 0.25 with Max–Abs normalization.

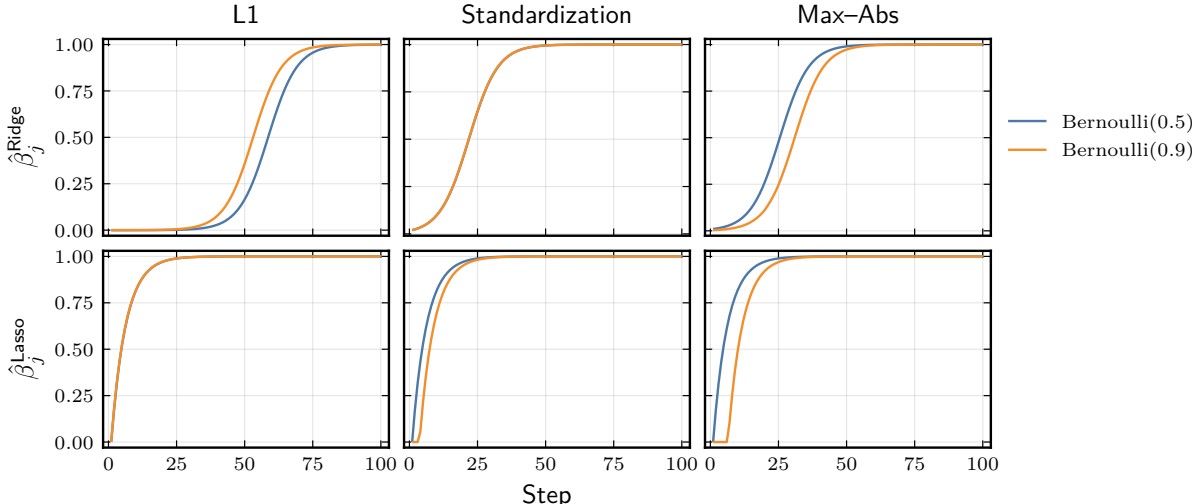

Figure 2: Unstandardized coefficients for lasso and ridge regression along a regularization path. The models are fit to an experiment with two binary features, generated to have class balances 0.5 and 0.9, respectively. We have generated the response as $y_i = \boldsymbol{x}_i^\intercal \boldsymbol{\beta}$ (without noise), with $\boldsymbol{\beta} = \mathbf{1}$. Note that the coefficients' paths overlap in for the L1 and Standardization normalization types in the lasso and ridge cases, respectively.

Our key contributions are:

1. We reveal that class balance in binary features directly affects lasso, ridge, and elastic net estimates, and show that scaling binary features with standard deviation (ridge) or variance (lasso) mitigates these effects at the cost of increased variance. Through extensive empirical analysis, we show that this finding extends to a wide range of settings (Section 4).

2. In our theoretical work, we examine this relationship in detail, showing that this bias from class imbalance holds even in the case of orthogonal features (Section 3.1). For the elastic net, however, we show that this effect *cannot* be mitigated by normalization and instead must be dealt with by scaling the penalty weights rather than the features (Section 3.4).

3. For mixed data designs, we demonstrate how normalization choices implicitly determine the relative regularization effects on binary versus continuous features (Section 3.2).

4. For interaction features, we show that a common alternative to normalizing interaction features leads to biased estimates and provide an alternative approach that mitigates this problem (Section 3.3).

Collectively, our results demonstrate that normalization is not merely a preprocessing step but rather an integral component of the model that requires careful consideration based on data characteristics and the chosen regularization approach.

## 2 Normalization

To avoid possible confusion regarding the ambiguous use of terminology in the literature, we will begin by clarifying what we mean by *normalization*, which we define as the process of centering and scaling the feature matrix.

**Definition 1** (Normalization). Let $X \in \mathbb{R}^{n \times p}$ be the feature matrix and let $c \in \mathbb{R}^p$ and $s \in \mathbb{R}^p_+$ be centering and scaling factors respectively. Then $\tilde{X}$ is the *normalized* feature matrix with elements given by $\tilde{x}_{ij} = (x_{ij} - c_j)/s_j$, where $i$ and $j$ index rows and columns, respectively.

Some authors refer to the procedure in Definition 1 as *standardization*, but here we define standardization only as the case when centering with the mean and scaling with the (uncorrected[2]) standard deviation.

There are many different normalization strategies and we have listed a few common choices in Table 2. Standardization is perhaps the most popular type of normalization, at least in the field of statistics. One of its benefits is that it simplifies certain aspects of fitting the model, such as fitting the intercept. The downside of standardization is that it involves centering by the mean, which destroys sparsity in $X$ since centering shifts zero values to non-zero.

When $X$ is sparse, two common alternatives to standardization are min–max and max–abs (maximum absolute value) normalization, which scale the data to lie in $[0, 1]$ and $[-1, 1]$ respectively, and therefore retain sparsity when features are binary. These methods are, however, both sensitive to outliers. And since sample extreme values often depend on sample size, as in the case of normal data (Section B.1), use of these methods may sometimes be problematic. Another alternative is to replace the $\ell_2$-norm in the standardization method with the $\ell_1$-norm, which leads to $\ell_1$-normalization. We note here that this method is often used without centering, but this would make the method depend on the mean of the feature, as in the case of max–abs normalization, and we therefore prefer the centered version here.

We have also included robust normalization in Table 2, which is a version of normalization that uses the median and interquartile range (IQR) as centering and scaling factors. Finally, we have also included the adaptive lasso (Zou, 2006), which is a special case of normalization that fits a standard ordinary least-squares regression (OLS) model to the data and uses the OLS estimates as scaling factors.[3] We have both of these methods in Table 2 for completeness, but we will not study it further in this paper.

---

[2]Standard deviation computed without Bessel's correction (use of $n-1$ instead of $n$ in the standard deviation formula).
[3]In the case when $p \gg n$, a ridge estimator is typically used instead.

Table 2: Common ways to normalize a matrix of features using centering and scaling factors $c_j$ and $s_j$, respectively. Note that $\bar{x}_j$ is the arithmetic mean of feature $j$ and that $Q_a(\boldsymbol{x}_j)$ is the $a$th quartile of feature $j$.

| Normalization | $c_j$ | $s_j$ |
|---|---|---|
| Standardization | $\bar{x}_j$ | $\frac{1}{\sqrt{n}}\|\boldsymbol{x}_j - \bar{x}_j\|_2$ |
| $\ell_1$-Normalization | $\bar{x}_j$ | $\frac{1}{\sqrt{n}}\|\boldsymbol{x}_j - \bar{x}_j\|_1$ |
| Max–Abs | $0$ | $\max_i |x_{ij}|$ |
| Min–Max | $\min_i(x_{ij})$ | $\max_i(x_{ij}) - \min_i(x_{ij})$ |
| Robust Normalization | $Q_2(\boldsymbol{x}_j)$ | $Q_3(\boldsymbol{x}_j) - Q_1(\boldsymbol{x}_j)$ |
| Adaptive Lasso | $0$ | $|\hat{\beta}_j^{\text{OLS}}|$ |

In the next section, we will examine how the choice of normalization affects the estimates for the lasso, ridge, and elastic net regression.

## 3 Ridge, Lasso, and Elastic Net Regression

In this setting we begin to describe the connection between normalization and the elastic net estimator. We start by showing how the scaling and centering parameters of the normalization method factors into the elastic net estimator in a general case. We then narrow our focus to the case of binary features and present our main results on bias, variance, and selection probability when the features are unbalanced.

Throughout the paper we assume that the response $\boldsymbol{y}$ is generated according to $\boldsymbol{y} = \beta_0^* + \boldsymbol{X}\boldsymbol{\beta}^* + \boldsymbol{\varepsilon}$, with $\boldsymbol{X}$ being the $n \times p$ design matrix with features (columns) $\boldsymbol{x}_j$, $\boldsymbol{\varepsilon}$ the vector of noise, with mean zero, finite variance $\sigma_\varepsilon^2$, and identically and independently distributed entries. We also assume $\boldsymbol{X}$, $\beta_0^*$, and $\boldsymbol{\beta}^*$ to be fixed and the features of the normalized design matrix to be orthogonal, that is, $\tilde{\boldsymbol{X}}^\intercal \tilde{\boldsymbol{X}} = \text{diag}\left(\tilde{\boldsymbol{x}}_1^\intercal \tilde{\boldsymbol{x}}_1, \ldots, \tilde{\boldsymbol{x}}_p^\intercal \tilde{\boldsymbol{x}}_p\right)$. In this case, it is a well-known fact (Tibshirani, 1996) that the solution to the elastic net problem is given by

$$\hat{\beta}_j^{(n)} = \frac{\text{S}_{\lambda_1}\left(\tilde{\boldsymbol{x}}_j^\intercal \boldsymbol{y}\right)}{\tilde{\boldsymbol{x}}_j^\intercal \tilde{\boldsymbol{x}}_j + \lambda_2}, \qquad \hat{\beta}_0^{(n)} = \frac{\boldsymbol{y}^\intercal \mathbf{1}}{n}, \tag{3}$$

where $\text{S}_\lambda(z)$ is the soft-thresholding operator, defined as $\text{S}_\lambda(z) = \text{sign}(z)\max(|z| - \lambda, 0)$, which is the proximal operator of the $\ell_1$ norm. We refer to Section B.2 for a derivation of the results above.

The assumption of orthogonal features may seem strong and is indeed almost never realised in practice. Here we use it for our theoretical results to show the direct connection between normalization and the elastic net estimator, and prove that even in this simple case, normalization has a pronounced effect on the estimates. In our experimental work (Section 4), however, we show that our findings extend to a much wider class of designs and also refer the reader to Section A, where we discuss this assumption in detail and provide additional theoretical and empirical results in relation to this.

Normalization changes the optimization problem and the estimated coefficients, which will now be on the scale of the normalized features. But here we are interested in $\hat{\boldsymbol{\beta}}$: the coefficients on the scale of the original problem. To obtain these, we transform the coefficients from the normalized problem, $\hat{\beta}_j^{(n)}$, back via $\hat{\beta}_j = \hat{\beta}_j^{(n)}/s_j$ for $j \in [p]$, where $[p] = \{1, 2, \ldots, p\}$. There is a similar transformation for the intercept, but we omit it here since we are not interested in interpreting it.

Taken together, this means that the solution for $\hat{\boldsymbol{\beta}}$ can be expressed as

$$\hat{\beta}_j = \frac{\text{S}_{\lambda_1}(\tilde{\boldsymbol{x}}_j^\intercal \boldsymbol{y})}{d_j}$$

where

$$\tilde{\boldsymbol{x}}_j^\mathsf{T}\boldsymbol{y} = \frac{\beta_j^* n\nu_j - \boldsymbol{x}_j^\mathsf{T}\boldsymbol{\varepsilon}}{s_j}$$

$$d_j = s_j\left(\frac{n\nu_j}{s_j^2} + \lambda_2\right) = s_j(\tilde{\boldsymbol{x}}_j^\mathsf{T}\tilde{\boldsymbol{x}}_j + \lambda_2),$$

(4)

with $\nu_j$ being the uncorrected sample variance of $\boldsymbol{x}_j$. The bias and variance of $\hat{\beta}_j$ are then given by

$$\mathrm{E}\,\hat{\beta}_j - \beta_j^* = \frac{1}{d_j}\,\mathrm{E}\,\mathrm{S}_\lambda(\tilde{\boldsymbol{x}}_j^\mathsf{T}\boldsymbol{y}) - \beta_j^*,$$

(5)

$$\mathrm{Var}\,\hat{\beta}_j = \frac{1}{d_j^2}\,\mathrm{Var}\,\mathrm{S}_\lambda(\tilde{\boldsymbol{x}}_j^\mathsf{T}\boldsymbol{y}).$$

(6)

See Section B.3 for a derivation of the results above as well as expressions for $\mathrm{E}\,\mathrm{S}_\lambda(x)$ and $\mathrm{Var}\,\mathrm{S}_\lambda(x)$.

These results hold in a general case. From now on, however, we will narrow our scope and assume that the entries of $\boldsymbol{\varepsilon}$ are identically, independently, and normally distributed, in which case both the bias and variance of $\hat{\beta}_j$ have analytical expressions (Section B.3.1) and

$$\tilde{\boldsymbol{x}}_j^\mathsf{T}\boldsymbol{y} \sim \mathrm{Normal}\left(\mu_j = \tilde{\boldsymbol{x}}_j^\mathsf{T}\boldsymbol{x}_j\beta_j^*, \sigma_j^2 = \tilde{\boldsymbol{x}}_j^\mathsf{T}\tilde{\boldsymbol{x}}_j\sigma_\varepsilon^2\right).$$

So far, we have assumed nothing about the features themselves, apart from being orthogonal to each other. The main focus of our paper, however, is binary features, which we will now turn to.

### 3.1  Binary Features

When $x_{ij} \in \{0,1\}$ for all $i$, we define $\boldsymbol{x}_j$ to be a *binary feature*, and the *class balance* of this feature as $q_j = \frac{1}{n}\sum_{i=1}^n x_{ij}$: the proportion of ones. It would make no difference to the majority of our results if we were to swap the ones and zeros as long as an intercept is included, and "class balance" is then equivalent to the proportion of either. But in the case of interactions (Section 3.3), the choice does in fact matter.

If feature $j$ is binary then $\nu_j = (q_j - q_j^2)$ (the uncorrected sample variance for a binary feature), which in Equation (4) yields

$$\tilde{\boldsymbol{x}}_j^\mathsf{T}\boldsymbol{y} = \frac{\beta_j^* n(q_j - q_j^2) - \boldsymbol{x}_j^\mathsf{T}\boldsymbol{\varepsilon}}{s_j}, \qquad d_j = s_j\left(\frac{n(q_j - q_j^2)}{s_j^2} + \lambda_2\right),$$

and consequently

$$\mu_j = \frac{\beta_j^* n(q_j - q_j^2)}{s_j} \quad\text{and}\quad \sigma_j^2 = \frac{\sigma_\varepsilon^2 n(q_j - q_j^2)}{s_j^2}.$$

We obtain bias and variance of the estimator with respect to $q_j$ by inserting $\tilde{\boldsymbol{x}}_j^\mathsf{T}\boldsymbol{y}$ and $d_j$ into Equations (5) and (6).

The presence of the factor $q_j - q_j^2$ in $\mu_j$, $\sigma_j^2$, and $d_j$ indicates a link between class balance and the elastic net estimator and, moreover, that this relationship is mediated by the scaling factor $s_j$. To achieve some initial intuition for this relationship, consider the noiseless case ($\sigma_\varepsilon = 0$) in which we have

$$\hat{\beta}_j = \frac{\mathrm{S}_{\lambda_1}(\tilde{\boldsymbol{x}}_j^\mathsf{T}\boldsymbol{y})}{s_j\left(\tilde{\boldsymbol{x}}_j^\mathsf{T}\tilde{\boldsymbol{x}}_j + \lambda_2\right)} = \frac{\mathrm{S}_{\lambda_1}\left(\frac{\beta_j^* n(q_j - q_j^2)}{s_j}\right)}{s_j\left(\frac{n(q_j - q_j^2)}{s_j^2} + \lambda_2\right)}.$$

(7)

This expression shows that class balance ($q_j$) directly affects the estimator through the factor $q_j - q_j^2$ (the variance of the binary feature). Starting with the lasso ($\lambda_2 = 0$), observe that the soft-thresholding part of the

estimator (numerator) diminishes for values of $q_j$ close to 0 or 1 unless we use the scaling factor $s_j = q_j - q_j^2$, in which case Equation (7) simplifies to

$$\hat{\beta}_j = \frac{\mathrm{S}_{\lambda_1}\left(\frac{\beta_j^* n(q_j - q_j^2)}{q_j - q_j^2}\right)}{(q_j - q_j^2)\left(\frac{n(q_j - q_j^2)}{(q_j - q_j^2)^2} + \lambda_2\right)} = \frac{\mathrm{S}_{\lambda_1}(\beta_j^* n)}{n},$$

which is independent of $q_j$. For other choices of $s_j$, the soft-thresholding part of the estimator will be affected by class balance and also depend on the size of $\lambda_1$, with larger values of $\lambda_1$ leading to a larger effect of class balance.

Turning to the ridge case ($\lambda_1 = 0$), the soft-thresholding part simplifies to the identity function and the shrinkage instead comes from the presence of $\lambda_2$ in the denominator, which will scale the estimator towards zero as $q_j$ approaches 0 or 1. Contrary to the lasso case, when then instead need to take $s_j = (q_j - q_j^2)^{1/2}$, in which case the expression becomes

$$\hat{\beta}_j = \frac{\mathrm{S}_{\lambda_1}\left(\frac{\beta_j^* n(q_j - q_j^2)}{(q_j - q_j^2)^{1/2}}\right)}{(q_j - q_j^2)^{1/2}\left(\frac{n(q_j - q_j^2)}{(q_j - q_j^2)} + \lambda_2\right)} = \frac{\beta_j^* n}{n + \lambda_2},$$

which is again independent of $q_j$.

Observe, however, that for the elastic net ($\lambda_1 > 0, \lambda_2 > 0$), there exists no $s_j$ that can make the estimator independent of $q_j$. In other words, there is no type of normalization, at least under our parameterization, that is able to mitigate the class balance bias in this case. In Section 3.4, however, we will show how to tackle this issue for the elastic net by scaling the penalty weights. But for now we continue to study the case of normalization.

Based on the reasoning above, we will consider the scaling parameterization $s_j = (q_j - q_j^2)^\delta$, $\delta \geq 0$, which includes the cases that we are primarily interested in, namely $\delta = 0$ (no scaling, as in min–max and max–abs normalization), $\delta = 1/2$ (standard-deviation scaling), and $\delta = 1$ (variance scaling). The last of these, variance scaling, is in fact equivalent to scaling with the mean-centered $\ell_1$-norm in this particular case of binary features. In Section B.4, we expand Equation (7) under this parameterization to clarify how the choice of $\delta$ affects class-balance bias in the lasso and ridge cases and why it is impossible (under our parameterization) to remove this bias in the case of the elastic net.

Another consequence of Equation (7), which holds also in the noisy situation, is that normalization affects the estimator even when the binary feature is balanced ($q_j = 1/2$). $\delta = 0$, for instance, scales $\beta_j^*$ in the input to $\mathrm{S}_\lambda$ by $n(q_j - q_j^2) = n/4$. $\delta = 1$, in contrast, imposes no such scaling in the class-balanced case. And for $\delta = 1/2$, the scaling factor is $n/2$. Generalizing this, we see that to achieve equivalent scaling in the class-balanced case for all types of normalization, under our parameterization, we would need to use $s_j = 4^{\delta-1}(q_j - q_j^2)^\delta$. But this only resolves the issue for the lasso. To achieve a similar effect for ridge regression, we would need another (but similar) modification. When all features are binary, we can just scale $\lambda_1$ and $\lambda_2$ to account for this effect,[4] which is equivalent to modifying $s_j$. But when we consider mixes of binary and normal features in Section 3.2, we need to exert extra care.

We now proceed to consider how class balance affects the bias, variance, and selection probability of the elastic net estimator under the presence of noise. A consequence of our assumption of a normal error distribution and consequent normal distribution of $\tilde{x}_j^\intercal y$ is that the probability of selection in the elastic net problem is given by

$$\Pr\left(\hat{\beta}_j \neq 0\right) = \Phi\left(\frac{\beta_j^* n(q_j - q_j^2)^{1/2} - \lambda_1(q_j - q_j^2)^{\delta-1/2}}{\sigma_\varepsilon \sqrt{n}}\right)$$
$$+ \Phi\left(\frac{-\beta_j^* n(q_j - q_j^2)^{1/2} - \lambda_1(q_j - q_j^2)^{\delta-1/2}}{\sigma_\varepsilon \sqrt{n}}\right). \tag{8}$$

---

[4]We use this strategy in all of the following examples.

where $\Phi$ is the cumulative distribution function of the standard normal distribution. Letting $\theta_j = -\mu_j - \lambda_1$ and $\gamma_j = \mu_j - \lambda_1$, we can express this probability asymptotically as $q_j \to 1^-$ as

$$\lim_{q_j \to 1^-} \Pr(\hat{\beta}_j \neq 0) = \begin{cases} 0 & \text{if } 0 \leq \delta < \frac{1}{2}, \\ 2\,\Phi\left(-\frac{\lambda_1}{\sigma_\varepsilon \sqrt{n}}\right) & \text{if } \delta = \frac{1}{2}, \\ 1 & \text{if } \delta > \frac{1}{2}. \end{cases} \tag{9}$$

In Figure 3, we plot this probability for various settings of $\delta$ for a single feature. Our intuition from the noiseless case holds: suitable choices of $\delta$ can mitigate the influence of class imbalance on selection probability. The lower the value of $\delta$, the larger the effect of class imbalance becomes. Note that the probability of selection initially decreases also in the case when $\delta \geq 1$. This is a consequence of increased variance of $\tilde{x}_j^\intercal y$ due to the scaling factor that inflates the noise term. But as $q_j$ approaches 1, the probability eventually rises towards 1 for $\delta \in \{1, 1.5\}$. The reason for this is that this rise in variance eventually quells the soft-thresholding effect altogether. Note, also, that the selection probability is unaffected by $\lambda_2$.

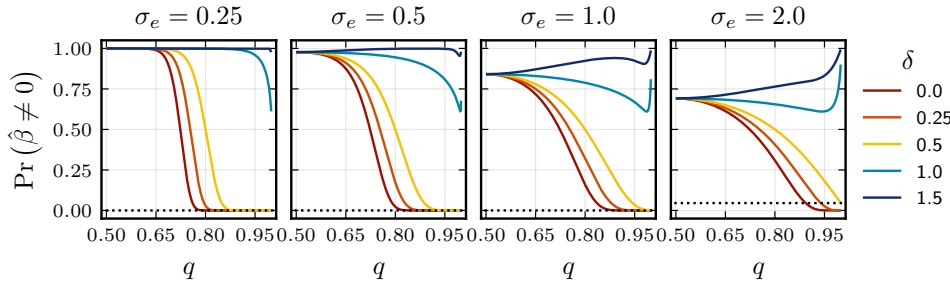

Figure 3: Probability of selection in the lasso given measurement noise $\sigma_\varepsilon$, regularization level $\lambda_1$, and class balance $q$. The scaling factor is set to $s_j = (q - q^2)^\delta$, $\delta \geq 0$. The dotted line represents the asymptotic limit for $\delta = 1/2$ from Equation (9).

Now we turn to the impact of class imbalance on bias and variance of the elastic net estimator. We begin, in Theorem 3.1, by considering the expected value of the elastic net estimator in the limit as $q_j \to 1^-$.

**Theorem 3.1.** *If $x_j$ is a binary feature with class balance $q_j \in (0, 1)$, $\lambda_1 \in [0, \infty)$, $\lambda_2 \in [0, \infty)$, $\sigma_\varepsilon > 0$, and $s_j = (q_j - q_j^2)^\delta$, $\delta \geq 0$ then*

$$\lim_{q_j \to 1^-} \mathrm{E}\,\hat{\beta}_j = \begin{cases} 0 & \text{if } 0 \leq \delta < \frac{1}{2}, \\ \frac{2n\beta_j^*}{n+\lambda_2}\,\Phi\left(-\frac{\lambda_1}{\sigma_\varepsilon \sqrt{n}}\right) & \text{if } \delta = \frac{1}{2}, \\ \beta_j^* & \text{if } \delta > \frac{1}{2}. \end{cases}$$

Theorem 3.1 shows that bias of the elastic net estimator approaches $-\beta_j^*$ as $q_j \to 1^-$ when $0 \leq \delta < 1/2$. When $\delta = 1/2$ (standardization), the estimate approaches a constant that depends on regularization strength, noise level, and the true strength of the coefficient. For $\delta > 1/2$, the estimate is asymptotically unbiased as a by-product of variance dominating in the limit as $q_j \to 1^{-1}$, which suggests that variance-scaling ($\ell_1$-norm normalization) could be problematic in a scenario with much noise and highly imbalanced features.

In Theorem 3.2, we continue by studying the variance in the limit as $q_j \to 1^-$, which shows that the variance of the elastic net estimator tends to $\infty$ in the limit unless the scaling parameter $s_j < 1/2$.

**Theorem 3.2.** *Assume the conditions of Theorem 3.1 hold, except that $\lambda_1 > 0$. Then*

$$\lim_{q_j \to 1^-} \mathrm{Var}\,\hat{\beta}_j = \begin{cases} 0 & \text{if } 0 \leq \delta < \frac{1}{2}, \\ \infty & \text{if } \delta \geq \frac{1}{2}. \end{cases}$$

Note that Theorem 3.2 applies only to the case when $\lambda_1 > 1$. In Theorem B.2 (Section B.5), we state the corresponding result for ridge regression.

Taken together, Theorems B.2, 3.1 and 3.2, indicate that the choice of scaling parameter constitutes a bias–variance trade-off with respect to $\delta$: increasing $\delta$ reduces class-balance bias, but does so at the cost of increased variance.

In Figure 4, we now visualize bias, variance, and mean-squared error for ranges of class balance and various noise-level settings for a lasso problem. The figure demonstrates the bias–variance trade-off that our asymptotic results suggest and indicates that the optimal choice of $\delta$ is related to the noise level in the data. Since this level is typically unknown and can only be reliably estimated in the low-dimensional setting, it suggests there might be value in selecting $\delta$ through hyper-parameter optimization.[5] In Figure 14 (Section B.7) we show results for ridge regression as well. As expected, it is then $\delta = 1/2$ that leads to unbiased estimates.

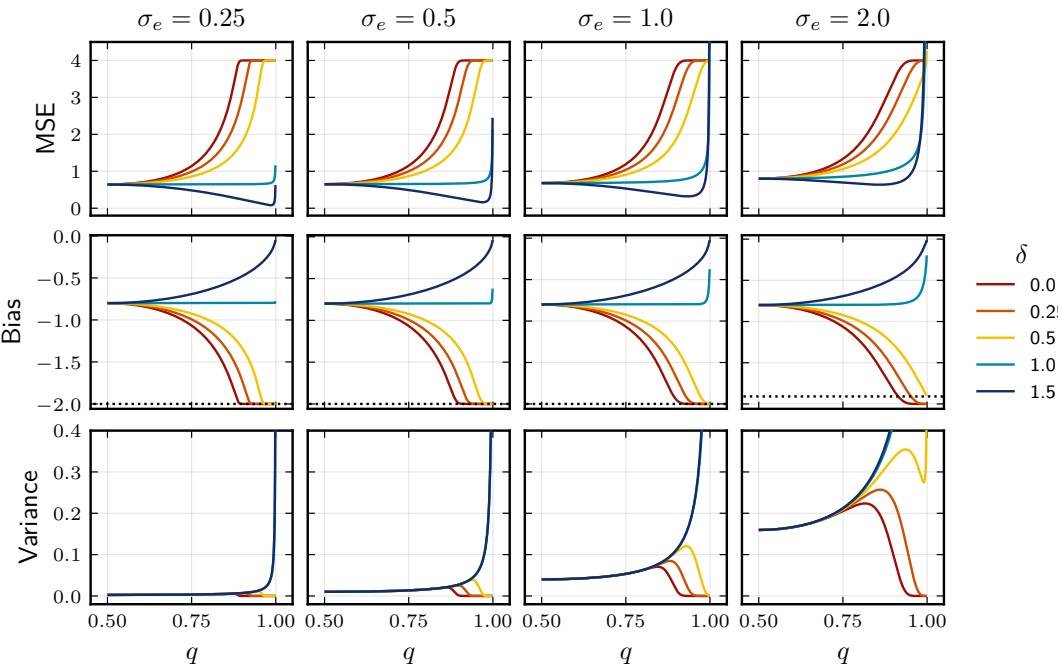

Figure 4: Bias, variance, and mean-squared error for a one-dimensional lasso problem, parameterized by noise level ($\sigma_\varepsilon$), class balance ($q$), and scaling ($\delta$). Dotted lines represent asymptotic bias of the lasso estimator in the case when $\delta = 1/2$.

So far, we have only considered a single binary feature, but in Section E.1 we present results on power and false discovery rates for problems with multiple features. In the next section we will also step beyond the all-binary context and consider mixes of binary and continuous features.

## 3.2 Mixed Data

A fundamental problem with mixes of binary and continuous features is deciding how to put these features on the same scale in order to regularize each type of feature fairly. In principle, we need to match a one-unit change in the binary feature with some amount of change in the normal feature. This problem has previously been tackled, albeit from a different angle, by Gelman (2008), who argued that the common default choice of presenting standardized regression coefficients unduly emphasizes coefficients from continuous features.

To setup this situation formally, we will say that the effects of a binary feature $\boldsymbol{x}_1$ and a normal feature $\boldsymbol{x}_2$ are *comparable* if $\beta_1^* = \kappa \sigma \beta_2^*$, where $\kappa > 0$ represents the number of standard deviations of the normal feature we consider to be comparable to one unit on the binary feature. As an example, assume $\kappa = 2$. Then, if $\boldsymbol{x}_2$ is sampled from Normal $\left(\mu_j, \sigma^2 = (1/2)^2\right)$, the effects of $\boldsymbol{x}_1$ and $\boldsymbol{x}_2$ are comparable if $\beta_1^* = 2\sigma\beta_2^* = \beta_2^*$.

---

[5]In Section E.4 we demonstrate the utility of doing so.

The definition above refers to $\boldsymbol{\beta}^*$, but for our regularized estimates we need $\hat{\beta}_1 = \kappa\sigma\hat{\beta}_2$ to hold. If we assume that we are in a noiseless situation ($\sigma_\varepsilon = 0$), are standardizing the normal feature, and that, without loss of generality, $\bar{x}_1 = 0$, then we need the following equality to hold:

$$\hat{\beta}_1 = \kappa\sigma\hat{\beta}_2 \implies \frac{\mathrm{S}_{\lambda_1}(\tilde{\boldsymbol{x}}_1^\mathsf{T}\boldsymbol{y})}{s_1\left(\tilde{\boldsymbol{x}}_1^\mathsf{T}\tilde{\boldsymbol{x}}_1 + \lambda_2\right)} = \frac{\kappa\sigma\,\mathrm{S}_{\lambda_1}(\tilde{\boldsymbol{x}}_2^\mathsf{T}\boldsymbol{y})}{s_2\left(\tilde{\boldsymbol{x}}_2^\mathsf{T}\tilde{\boldsymbol{x}}_2 + \lambda_2\right)} \implies \frac{\mathrm{S}_{\lambda_1}\left(\frac{n\beta_1^*(q-q^2)}{s_1}\right)}{s_1\left(\frac{n(q-q^2)}{s_1^2} + \lambda_2\right)} = \frac{\kappa\,\mathrm{S}_{\lambda_1}\left(\frac{n\beta_1^*}{\kappa}\right)}{n + \lambda_2}. \tag{10}$$

For the lasso ($\lambda_2 = 0$) and ridge regression ($\lambda_1 = 0$), we see that the equation holds for $s_1 = \kappa(q - q^2)$ and $s_1 = (q - q^2)^{1/2}$, respectively. In other words, we achieve comparability in the lasso by scaling each binary feature with its variance times $\kappa$. And for ridge regression, we can achieve comparability by scaling with standard deviation, irrespective of $\kappa$. For any other choices of $s_1$, equality holds only at a fixed level of class balance. Let this level be $q_0$. Then, to achieve equality for $\lambda_2 = 0$, we need $s_1 = \kappa(q_0 - q_0^2)^{1-\delta}(q - q^2)^\delta$. Similarly, for $\lambda_1 = 0$, we need $s_1 = (q_0 - q_0^2)^{1-2\delta}(q - q^2)^\delta$. In the sequel, we will assume that $q_0 = 1/2$, to have effects be equivalent for the class-balanced case.

Note that this means that the choice of normalization has an implicit effect on the relative penalization of binary and normal features—even in the class-balanced case ($q_1 = 1/2$). If we for instance use $\delta = 0$ and fit the lasso, then Equation (10) for a binary feature with $q_1 = 1/2$ becomes $4\,\mathrm{S}_{\lambda_1}\left(n\beta_1^*/4\right) = \kappa\,\mathrm{S}_{\lambda_1}(n\beta_1^*/\kappa)$, which implies $\kappa = 4$. In other words, the choice of normalization equips our model with a belief about how binary and normal features should be penalized relative to one another.

For the rest of this paper, we will use $\kappa = 2$ and say that the effects are comparable if the effect of a flip in the binary feature equals the effect of a two-standard deviation change in the normal feature. We motivate this by an argument by Gelman (2008), but want to stress that the choice of $\kappa$ should, if possible, be based on contextual knowledge of the data and that our results depend only superficially on this particular setting.

### 3.3 Interactions

The elastic net can be extended to include interactions. There is previous literature on this topic (Bien et al., 2013; Lim & Hastie, 2015; Zemlianskaia et al., 2022), but it has not considered the possible influence of normalization. Here, we will consider simple pairwise interactions with no restriction on the presence of main effects. For our analysis, we let $\boldsymbol{x}_1$ and $\boldsymbol{x}_2$ be two features of the data and $\boldsymbol{x}_3$ their interaction, so that $\beta_3$ represents the interaction effect.

We consider two cases in which we assume that the features are orthogonal and that $\boldsymbol{x}_1$ is binary with class balance $q_1$. In the first case, we let $\boldsymbol{x}_2$ be normal with mean $\mu$ and variance $\sigma^2$, and in the second case $\boldsymbol{x}_2$ be binary with class balance $q_2$. To construct the interaction feature, we center[6] the main features and then multiply element-wise. The elements of the interaction feature are then given by $x_{3,i} = (x_{1,i} - \bar{\boldsymbol{x}}_1)(x_{2,i} - \bar{\boldsymbol{x}}_2)$.

If $\boldsymbol{x}_2$ is normal and both features are centered before computing the interaction term, the variance becomes $\sigma^2(q - q^2)$, which suggests using $s_3 = \sigma(q - q^2)^\delta$ along the lines of our previous reasoning. And if $\boldsymbol{x}_2$ is binary, instead, then similar reasoning suggests using $s_3 = ((q_1 - q_1^2)(q_2 - q_2^2))^\delta$. In Section 4.1.4, we study the effects of these choices in simulated experiments.

### 3.4 The Weighted Elastic Net

We have so far shown that certain choices of normalization can mitigate the class-balance bias imposed by the lasso and ridge regularization. But we have also demonstrated (Section 3.1) that there is no (simple) choice of scaling that can achieve the same effect for the elastic net. Equation (7), however, suggests a natural alternative to normalization, which is to use the weighted elastic net, in which we minimize

$$\frac{1}{2}\|\boldsymbol{y} - \beta_0 - \boldsymbol{X}\boldsymbol{\beta}\|_2^2 + \lambda_1 \sum_{j=1}^p u_j|\beta_j| + \frac{\lambda_2}{2} \sum_{j=1}^p v_j\beta_j^2,$$

---

[6] See Section B.6 for motivation for why we center the features before computing the interaction.

with $\boldsymbol{u}$ and $\boldsymbol{v}$ being $p$-length vectors of positive scaling factors. This is equivalent to the standard elastic net for a normalized feature matrix when $u_j = s_j$ and $v_j = s_j^2$, which can be seen by substituting $\beta_j s_j = \tilde{\beta}_j$ in Equation (2) and solving for $\tilde{\boldsymbol{\beta}}$. Note that we do not need to rescale the coefficients from this problem as we would for the standard elastic net on normalized data.

This allows us to control class-balance bias by setting our weights according to $u_j = v_j = (q_j - q_j^2)^\omega$ and counteract it, at least in the noiseless case, with $\omega = 1$, which, we want to emphasize, is *not* possible using the standard elastic net. For the lasso and ridge regression, however, this setting of $\omega = 1$ is equivalent to using $\delta = 1$ and $\delta = 1/2$, respectively, in the standard elastic net with normalized data. Results analogous to those in Section 3.1 can be attained with a few small modifications for the weighted elastic net case. Starting with selection probability, we can set $s_j = 1$ and replace $\lambda_1$ with $\lambda_1 u_j = \lambda_1 (q_j - q_j^2)^\omega$ in Equation (8), which shows that $\omega$ and $\delta$ have interchangeable effects for selection probability.

As far as expected value and variance of the weighted elastic net estimator is concerned, the same expressions apply directly in the case of the weighted elastic net given $s_j = 1$ for all $j$ and replacing $\lambda_1$ as in the previous paragraph and $\lambda_2$ with $\lambda_2 (q_j - q_j^2)^\omega$. On the other hand, the asymptotic results differ slightly as we now show.

**Theorem 3.3.** *Let $\boldsymbol{x}_j$ be a binary feature with class balance $q_j \in (0, 1)$ and take $\lambda_1 > 0$, $\lambda_2 > 0$, and $\sigma_\varepsilon > 0$. For the weighted elastic net with weights $u_j = v_j = (q_j - q_j^2)^\omega$ and $\omega \geq 0$, it holds that*

$$\lim_{q_j \to 1^-} \mathrm{E}\,\hat{\beta}_j = \begin{cases} 0 & \text{if } 0 \leq \omega < 1, \\ \frac{\beta^* n}{n + \lambda_2} & \text{if } \omega = 1, \\ \beta^* & \text{if } \omega > 1, \end{cases} \qquad \lim_{q_j \to 1^-} \mathrm{Var}\,\hat{\beta}_j = \begin{cases} 0 & \text{if } 0 \leq \omega < \frac{1}{2}, \\ \infty & \text{if } \omega \geq \frac{1}{2}. \end{cases}$$

This result for expected value is similar to the one for the unweighted but normalized elastic net. The only difference arises in the case when $\omega = 1$, in which case the limit is unaffected by $\lambda_1$ in the case of the weighted elastic net. For variance, the result mimics the result for the elastic net with normalization. The results for bias, variance, and mean-squared error for the weighted elastic net are similar to those in Figure 4 and are plotted in Figure 12 (Section B.7).

This wraps up our theoretical contributions in our paper. In the coming section, we will turn to empirical experiments and demonstrate that our theoretical results both hold in practice and, at least empirically, extend beyond our current assumptions.

# 4 Experiments

In the following sections we present the results of our experiments. For all simulated data we generate our response vector according to $\boldsymbol{y} = \boldsymbol{X}\boldsymbol{\beta}^* + \boldsymbol{\varepsilon}$, with $\boldsymbol{\varepsilon} \sim \text{Normal}(\boldsymbol{0}, \sigma_\varepsilon^2 \boldsymbol{I})$. We consider two types of features: binary (quasi-Bernoulli) and quasi-normal features. To generate binary vectors, we sample $\lceil nq_j \rceil$ indexes uniformly at random without replacement from $[n]$ and set the corresponding elements to one and the remaining ones to zero. To generate quasi-normal features, we generate a linear sequence $\boldsymbol{w}$ with $n$ values from $10^{-4}$ to $1 - 10^{-4}$, set $x_{ij} = \Phi^{-1}(w_i)$, and then shuffle the elements of $\boldsymbol{x}_j$ uniformly at random.

We use a coordinate solver from Lasso.jl (Kornblith, 2024) to optimize our models, which we have based on the algorithm outlined by Friedman et al. (2010). All experiments were coded using the Julia programming language (Bezanson et al., 2017) and the code is available at https://github.com/jolars/normreg. All simulated experiments were run for at least 100 iterations and, unless stated otherwise, are presented as means $\pm$ one standard deviation (using bars or ribbons).

## 4.1 Normalization in Lasso and Ridge Regression

In this section we consider fitting the lasso and ridge regression to normalized datasets. To normalize the data, we standardize all quasi-normal features. For binary features, we center by mean and scale by $s_j \propto (q_j - q_j^2)^\delta$.

### 4.1.1 Variability and Bias in Estimates

In our first experiment, we consider fitting the lasso to a simulated dataset with $n = 500$ observations and $p = 1000$ features. The first 20 features correspond to signals, with $\beta_j^* = 1$, and otherwise we set $\beta_j^*$ to 0. Furthermore, we set the class balance of the first 20 features so that it increases geometrically from 0.5 to 0.99. For all other features we pick $q_j$ uniformly at random in $[0.5, 0.99]$. We estimate the regression coefficients using the lasso, setting $\lambda_1 = 2\sigma_\varepsilon\sqrt{2\log p}$, with $\sigma_\varepsilon$ set to achieve a signal-to-noise ratio (SNR) of 2. In addition, we introduce correlation between the features by copying the first $\lceil \rho n/2 \rceil$ values from the first feature to each of the following features.

The results (Figure 5) show that class balance has considerable effect, particularly in the case of no scaling ($\delta = 0$), which corroborates our results in Section 3.1. At $q_j = 0.99$, for instance, the estimate ($\hat{\beta}_{20}$) is consistently zero when $\delta = 0$. For larger values of $\delta$, we see that class imbalance leads to increased estimation variance in accordance with our theory for the orthogonal case. The effect of correlation appears to have no effect on this class-balance bias.

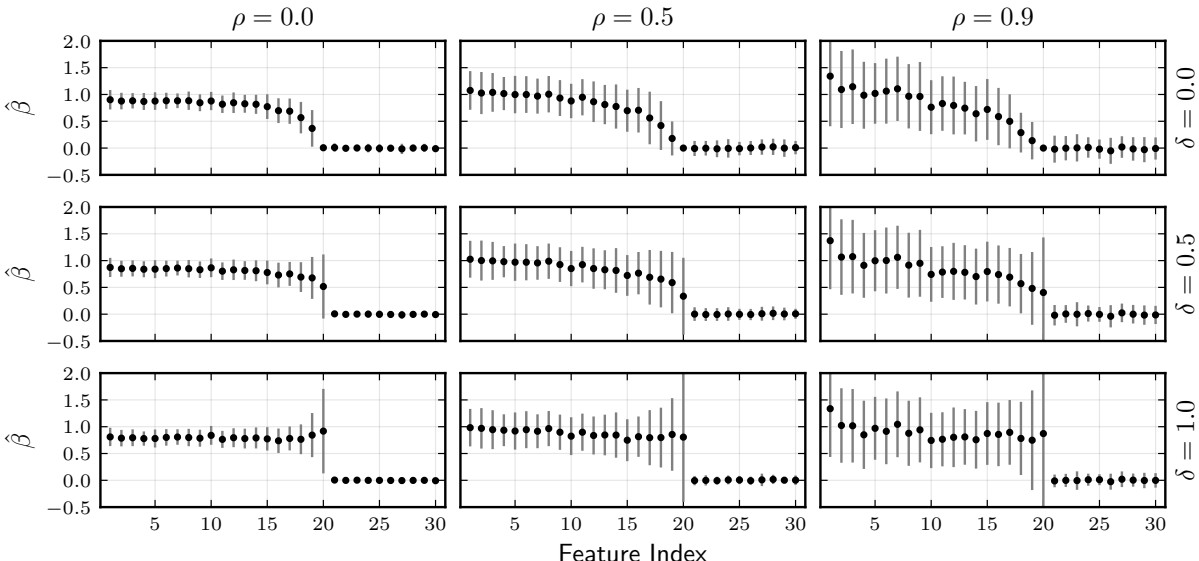

Figure 5: Regression coefficients for a lasso problem with binary data with $n = 500$ and $p = 1000$. For $j \in \{1, 2, \ldots, p\}$, we set $\beta_j^* = 1$ and let $q_j$ increase geometrically from 0.5 to 0.99. For the remaining features, we pick $q_j$ uniformly at random from $[0.5, 0.99]$ and set $\beta_j^* = 0$. We show only the first 30 coefficients.

### 4.1.2 Predictive Performance

In this section we examine the influence of normalization on predictive performance for three different datasets: `rhee2006` (Rhee et al., 2006), `eunite2001` (Chen et al., 2004), and `triazines` (Hirst et al., 1994; King et al., 1995).[7] We present the results for lasso and ridge regression in Figure 6, which shows contour plots of the validation set error in terms of normalized mean-squared error (NMSE). We see that optimal setting of $\delta$ differs between the different datasets: for `eunite2001`, both the lasso and ridge are quite insensitive to the type of normalization, and low error is attainable for the full range of $\delta$. For `rhee2006`, this holds for the lasso too, but not for ridge regression, where a value in approximately $[0, 3]$ is optimal. Finally, for `triazines`, the problem is quite sensitive to the choice of $\delta$ as well as the type of model used.

Also see Section E.2, where we show how the support size of the lasso solutions vary with $\delta$ and $\lambda$ and Section E.3, where we complement these results with experiments on simulated data under various class balances and signal-to-noise ratios, again showing that normalization has a strong impact on predictive performance.

---

[7]See Section G for details about these datasets.

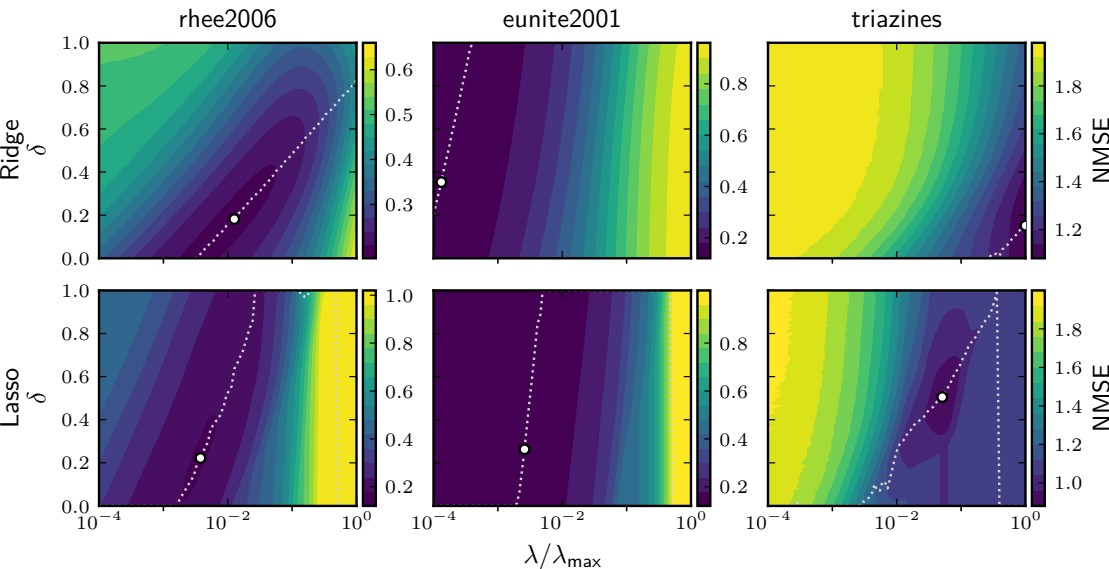

Figure 6: Contour plots of normalized validation set mean-squared error (NMSE) for $\delta$ and $\lambda$ in lasso and ridge regression on three real data sets: `rhee2006`, `eunite2001`, and `triazines`. The dotted path shows the smallest NMSE as a function of $\lambda$ and the circles mark combinations with the lowest error.

### 4.1.3   Mixed Data

In Section 3.2 we discussed the issue of normalizing mixed data. Here, we examine this issue empirically. We construct a quasi-normal feature with mean zero and standard deviation $1/2$ and a binary feature with varying class balance $q_j$. We set the signal-to-noise ratio to 0.5 and use $n = 1000$. These features are constructed so that their effects are comparable under the notion of comparability that we introduced in Section 3.2 using $\kappa = 2$. In order to preserve the comparability for the baseline case when we have perfect class balance, we scale by $s_j = 2 \times (1/4)^{1-\delta}(q_j - q_j^2)^\delta$. Finally, we set $\lambda$ to $\lambda_{\max}/2$ and $2\lambda_{\max}$ for lasso and ridge regression respectively.

The results (Figure 7) reflect our theoretical results from Section 3. In the case of the lasso, we need $\delta = 1$ (variance scaling) to avoid the effect of class imbalance, whereas for ridge we instead need $\delta = 1/2$ (standardization). As our theory suggests, this extra scaling mitigates this class-balance bias at the cost of added variance.

### 4.1.4   Interactions

Next, we study the effects of normalization and class balance on interactions in the lasso. Our example consists of a two-feature problem with an added interaction term given by $x_{i3} = x_{i1}x_{i2}$. The first feature is binary with class balance $q$ and the second quasi-normal with standard deviation 0.5. We use $n = 1000$, $\lambda_1 = n/4$, and normalize the binary feature by mean-centering and scaling by $\kappa(q - q^2)$, using $\kappa = 2$. We consider two different strategies for choosing $s_3$: in the first strategy, which we call *Strategy 1*, we simply standardize the resulting interaction feature. This is a common strategy used, for instance in Bien et al. (2013); Lim & Hastie (2015). In the second strategy, *Strategy 2*, we center with mean and scale with $s_1 s_2$ (the product of the scales of the binary and normal features).

The results (Figure 8) show that only Strategy 2 estimates the effect of the interaction correctly. Strategy 1, meanwhile, only selects the correct model if the class balance of the binary feature is close to $1/2$ and in general shrinks the coefficient too much.

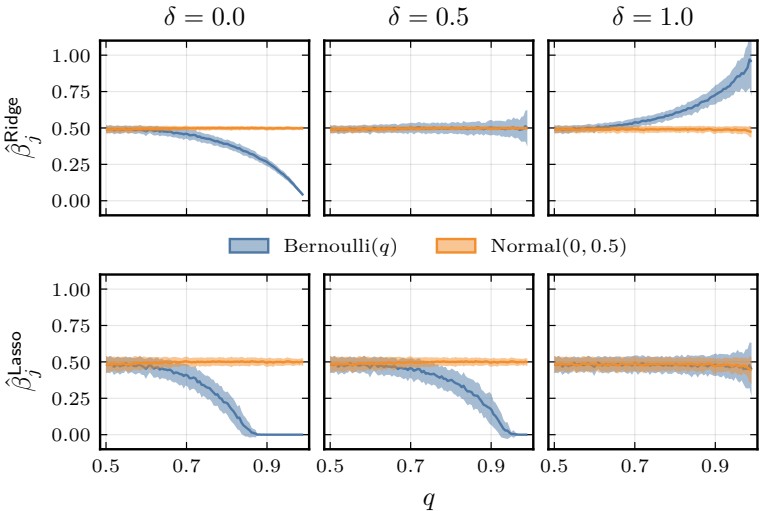

Figure 7: Lasso and ridge estimates for a two-dimensional problem where one feature is a binary feature with class balance $q_j$ (Bernoulli($q_j$)) and the other is quasi-normal with standard deviation $1/2$, (Normal$(0, 0.5)$).

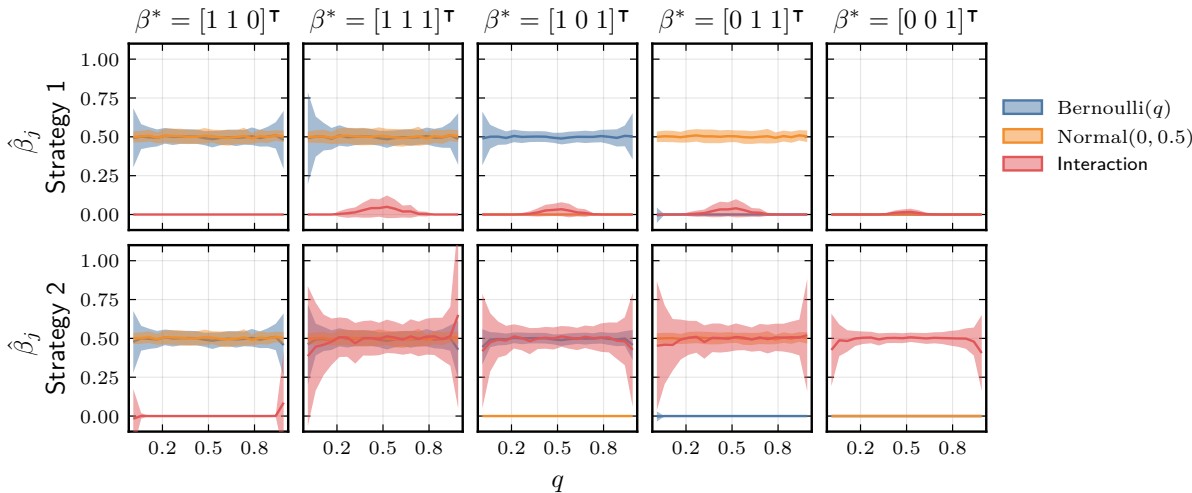

Figure 8: Lasso estimates for a problem with a binary feature, a quasi-normal feature, and an interaction feature. We have varied the true signal $\boldsymbol{\beta}^*$ and use two different normalization strategies for the interaction, where Strategy 1 represents standardization and Strategy 2 is mean-centering together with scaling by $s_1 s_2$.

## 4.2 The Weighted Elastic Net

The weighted elastic net can be used as an alternative to normalization to correct for class balance bias when $\lambda_1 > 0$ and $\lambda_2 > 0$. To simplify the presentation, we parameterize the elastic net as $\lambda_1 = \alpha\lambda$ and $\lambda_2 = (1 - \alpha)\lambda$, so that $\alpha$ controls the balance between the ridge and lasso. We conduct an experiment with the same setup as in Section 4.1.3, but here we use the weighted elastic net instead with $\alpha = 0.5$. We use $n = 1000$ and vary $\omega$, using the weights $u_j = v_j = (q_j - q_j^2)^\omega$ as we suggested in Section 3.4. Our results (Figure 9) show that $\omega = 1$ leads to seemingly unbiased estimates.

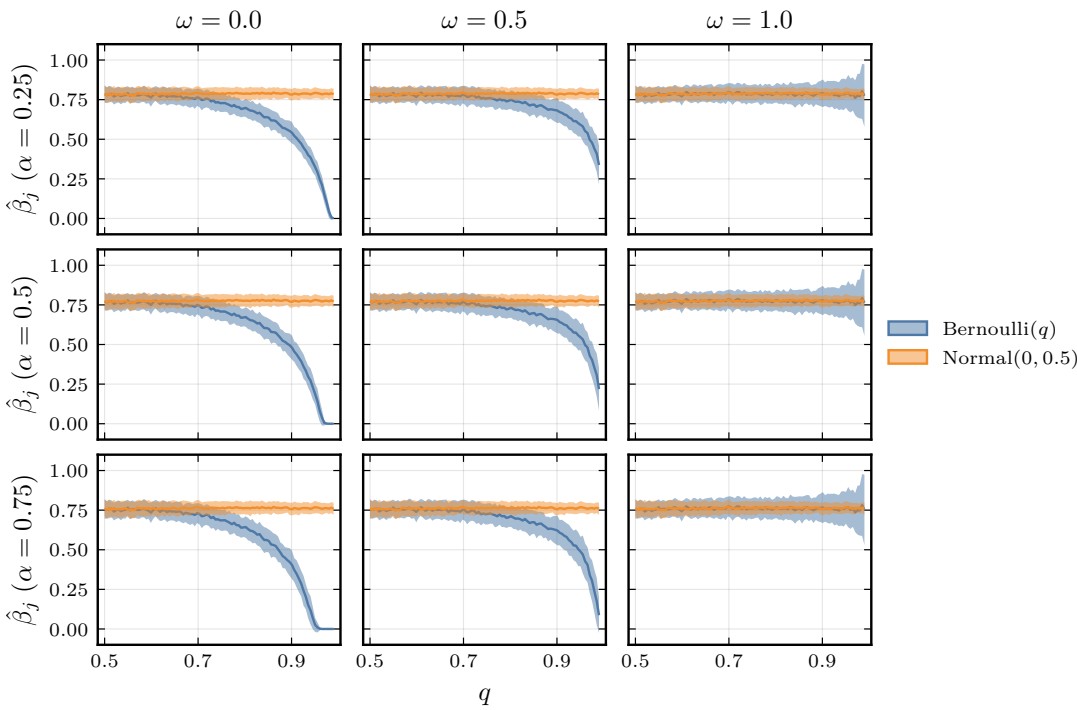

Figure 9: Weighted elastic net estimates for $\alpha = 0.5$ for a problem with a binary feature with class balance $q$ (Bernoulli($q$)) and quasi-normal with standard deviation $1/2$ (Normal$(0, 0.5)$). $\omega$ indicates the scaling of the penalty weights.

## 4.3 Additional Results

In Section E we present extended results of our experiments as well as several additional experiments. For instance, we also consider an experiment on the Boston housing dataset in Section E.5, where we try to estimate the effect of normalization on dichotomized versions of the features and find that $\delta = 1$ leads to feature ranks that best correspond to the linear regression solution.

## 5 Practical Recommendations

The results presented in this paper directly concern practitioners working with regularized regression on data with binary features. In general, we want to stress that the choice of normalization affects not only the computational aspects of fitting the model, but also the model itself. The bias introduced by ridge, lasso, and elastic net regression depends on the normalization scheme used, and does so in a penalty-specific manner, which means there exists no one-size-fits-all solution. In the following sections, we provide some concrete guidelines for practitioners on how to normalize.

### 5.1 Feature Selection

We have shown that, to avoid bias against class-imbalanced binary features, we need to scale them with their standard deviations for ridge regression and with their variances for the lasso. Therefore, if the goal is feature selection and the cost of missing a relevant feature is high, then scaling with variance may be preferable. Scaling with variance, however, will increase the variability of the estimates, which means that the choice of normalization is a bias–variance trade-off. Naturally, this also means that the size of the dataset matters: with more data, users may be able to afford the increased variance that comes with scaling with variance.

### 5.2 Prediction

If the goal is prediction, then scaling with standard deviation or even not scaling at all[8] will typically prove the better choice. If the practitioner can afford the computational cost, we also recommend treating normalization as a hyper-parameter optimization problem, which will allow users to look for a normalization scheme that is optimal for their specific dataset and prediction task. The user can, for instance, cross-validate over a few common normalization schemes or alternatively use the parameterization through $\delta$ that we have proposed in Section 3.1.

### 5.3 The Elastic Net

For ridge and lasso regression, the class imbalance can be controlled through the choice of normalization. This is not, however, the case for the elastic net. In this case, we therefore recommend users scale the penalty weights instead of the features themselves, as described in Section 3.4. We believe this is a more intuitive approach, given that the elastic net is a mix of two different norms, which interact with the features' scales in separate ways.

### 5.4 Mixed Data

For mixed data, the choice of normalization is tricky since it implicitly affects the relative penalization of binary and normal features. Practitioners need to decide how they want to treat binary features relative to normal features. We discuss this topic in Section 3.2 and introduce the parameter $\kappa$ that allows practitioners to control this effect through the normalization strategy. While we provide no concrete recommendations for setting this parameter, we believe that awareness of this implicit relationship is important.

### 5.5 Interactions

Interactions introduce another layer of complexity because normalization will affect the correlation between the original features and their interaction terms. We recommend always centering the features before creating the interaction terms, since the means of the features otherwise affect the estimates. Second, we suggest a new type of normalization for interaction features, which is to scale them based on the *product* of the scales of the normalized original features. We have shown that this can mitigate the class-balance bias effect.

## 6 Discussion

This is the first paper to study the effects of normalization in lasso, ridge, and elastic net regression with binary data. We have discovered that the class balance (proportion of ones) of these binary features has a pronounced effect on both lasso and ridge estimates and that this effect depends on the type of normalization used. For the lasso, for instance, features with large class imbalances stand little chance of being selected if the binary features are standardized or, to an even greater extent, if they are not scaled at all—even if their relationships with the response are strong. Not scaling binary features is a common approach in practice, for instance recommended in the scikit-learn documentation when the data is sparse (scikit-learn developers, 2025). As is standardization, which is the default in many software packages for the lasso, such as glmnet (Friedman et al., 2010), as well as the practice taken by countless applications in research. All

---

[8]Or equivalently, using max–abs or min–max normalization.

in all, this means that the class-balance bias effect is likely pervasive in practice and risks having already influenced the conclusions drawn in analyses of many datasets.

The driver of this bias is the relationship between the variance of the feature and type of normalization. This works as expected for normally distributed features. But for binary features it means that a one-unit change is treated differently depending on the corresponding feature's class balance, which we believe may surprise some. We have, however, shown that scaling binary features with standard deviation in the case of ridge regression and variance in the case of the lasso mitigates this effect, but that doing so comes at the price of increased variance. This effectively means that the choice of normalization constitutes a bias–variance trade-off.

We have also studied the case of mixed data: designs that include both binary and normally distributed features (Section 3.2). In this setting, our first finding is that there is an implicit relationship between the choice of normalization and the manner in which regularization affects binary vis-à-vis normally distributed features, even when the binary feature is perfectly balanced. The choice of max–abs normalization, for instance, leads to a specific weighting of the effects of binary features relative to those of normal features.

For interactions between binary and normal features (Section 4.1.4), our conclusion is that the interaction feature—contrary to what recent literature on interactions in the lasso recommends—should be scaled with the *product* of the standard deviation of the normal feature and variance of the binary feature to avoid this effect of class imbalance. We have not seen this recommendation in the literature before, but it is a natural extension of our other results.

We note that our theoretical results are limited by a few assumptions: 1) a fixed feature matrix $X$, 2) normal and independent errors, and 3) orthogonal features. The first and second of these assumptions are standard in the literature. The third assumption on orthogonality, however, is strong and rarely satisfied in practice. Yet, as we show in Section A and our experiments, the assumption does not in fact appear to be restrictive for our results, which, at least empirically, hold under much more general settings. We have also focused on the case of binary and continuous features here, but are convinced that categorical features are also of interest and might raise additional challenges with respect to normalization. Finally, most of our results are restricted to least-squares loss, but since all generalized linear models (GLMs) are parameterized by the linear predictor, which we have shown to depend directly on class balance, we believe that our results are also relevant for other loss functions. Our initial results in Section E.4 seem to support this claim, but we defer further investigation of this to future work.

Regularized regression models are widely used in practice, and are staples of popular machine learning and statistical software packages such as glmnet (Friedman et al., 2010), scikit-learn (Pedregosa et al., 2011), mlpack (Curtin et al., 2023), skglm (Bertrand et al., 2022), LIBLINEAR (Fan et al., 2008), and MATLAB (The MathWorks Inc., 2022). Our results suggest that the choice of normalization is an important aspect of using these models that, in spite of the popularity of these methods, has so far been overlooked. We hope that our results will motivate researchers and practitioners to consider the choice of normalization more carefully in the future.

### Acknowledgments

We are grateful to Mathurin Massias and Malgorzata Bogdan for valuable feedback on earlier drafts of this manuscript.

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

## A   Orthogonality Assumption

A key assumption in our theoretical results is that the features are orthogonal to one another. In general, this is a strong assumption that is rarely satisfied in practice. In the high-dimensional setting with binary data, it is even impossible to achieve. In spite of this, it turns out that the assumption holds little bearing on our results.

The primary reason for this is a well-known behavior of regularized estimators when features are correlated. Since information about the effect is shared between the correlated features, the objective can attain a lower value by favoring one of the features over the others. The effect is particularly strong in the lasso, which is known to select only one of the correlated features and ignore the others given a sufficiently large correlation and penalty strength.

A second reason for why the assumption is not as restrictive as it may seem is that the correlation between two features, at least one being binary, tends to zero as class balance increases towards one. We first demonstrate this empirically in the following experiment. We consider a setting with two binary features: the first, $x_1$

with class balance $q_1 = 0.5$, and the second, $\boldsymbol{x}_2$, with varying balance $q_2 \in [0.5, 0.9]$. We set the true effect to $\beta_1^* = \beta_2^* = 1$ and the level of correlation, $\rho$ to three different levels $(0, 0.4, \text{ and } 0.6)$. The noise level $\sigma_\varepsilon$ is set to obtain a signal-to-noise ratio of 1. We generate $n = 10\,000$ observations and fit the lasso with $\lambda = \lambda_{\max}/2$.

The results are shown in Figure 10 where we see that the effect of correlation has no impact on the shrinkage imposed from decreasing class balance of the second feature. The results in fact suggest that the effect of $q_2$ is *stronger* when the features are correlated, which is due to the nature of the lasso that we previously discussed.

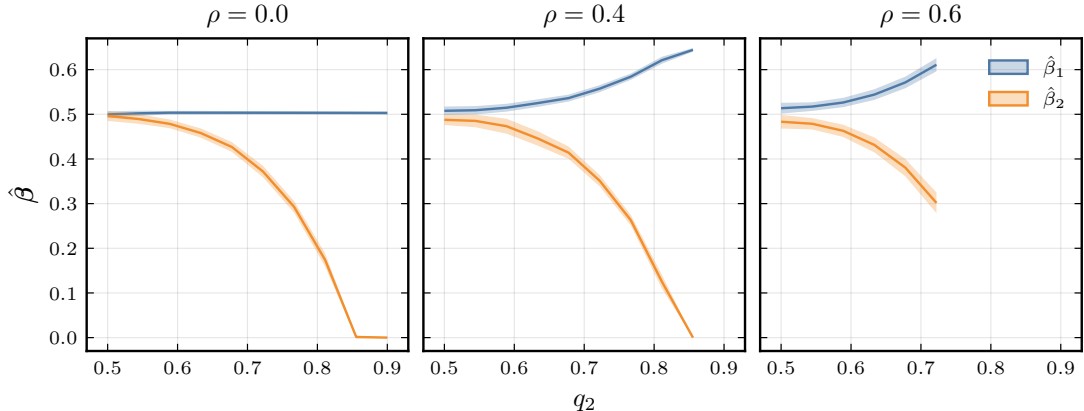

Figure 10: Estimates of the regression coefficients from the lasso, $\hat{\boldsymbol{\beta}}$, for the two features in the experiment in Section A. The first feature is binary with class balance $q_1 = 0.5$ and the second is binary with class balance $q_2 \in [0.5, 0.9]$. The features are correlated with correlation $\rho$. The plot shows means and 95% normal confidence intervals averaged over 100 iterations. Note that it's impossible to achieve high levels of correlation when the second feature is highly imbalanced, which is why the results for $\rho = 0.4$ and $0.6$ do not cover the whole range of $q_2$ values.

In the following theorems and corollary, we provide further evidence of this behavior by showing that the correlation between a binary feature and a continuous or binary feature tends to zero as the class balance increases.

In conclusion, although the orthogonality assumption is strong in general, unrealistic, and in many other areas of statistics make results hard to generalize to a wider setting, it does not seem to have a significant impact on our results.

**Theorem A.1** (Correlation for Dichotomized Normal Variables). *Let $X$ and $Y$ be two standard normal random variables with correlation $\rho$. Define $Z = \mathbf{1}[Y > \alpha]$ where $\alpha = \Phi^{-1}(q)$ is the quantile at which $Y$ is dichotomized. Then*

$$\mathrm{Corr}(X, Z) = \frac{\rho\,\phi(\alpha)}{\sqrt{q(1-q)}} \to 0 \quad as \quad q \to 1$$

*Proof.* Let $Z = \mathbf{1}[Y > \alpha]$ with $\alpha = \Phi^{-1}(q)$. Then, using the law of total expectation, we have

$$\mathrm{E}(XZ) = \mathrm{E}\big(\mathrm{E}(XZ \mid Y)\big) = \mathrm{E}\big(\rho Y \mathbf{1}[Y > \alpha]\big) = \rho \int_\alpha^\infty y\,\phi(y)\,\mathrm{d}y = \rho\,\phi(\alpha).$$

Since $\mathrm{Var}(X) = 1$ and $\mathrm{Var}(Z) = q(1-q)$, it follows that

$$\mathrm{Corr}(X, Z) = \frac{\mathrm{E}(XZ)}{\sqrt{\mathrm{Var}(X)\,\mathrm{Var}(Z)}} = \frac{\rho\,\phi(\alpha)}{\sqrt{q(1-q)}}.$$

And since $\phi(\alpha) \to 0$ exponentially fast as $q \to 1$, it follows that $\mathrm{Corr}(X, Z) \to 0$ as $q \to 1$. $\qquad\square$

**Theorem A.2** (Correlation with Bernoulli Variable). *Let $X$ be a continuous random variable and $Y$ be a Bernoulli random variable defined as:*

$$Y = \begin{cases} 1 & \text{with probability } p, \\ 0 & \text{with probability } 1 - p. \end{cases}$$

*Let $\mu_1 = \mathrm{E}\left(X \mid Y = 1\right)$, $\mu_0 = \mathrm{E}\left(X \mid Y = 0\right)$ and $\sigma_X^2 = V\left(X\right)$ Then the correlation between $X$ and $Y$ is given by:*

$$\rho_{X,Y} = \frac{(\mu_1 - \mu_0)}{\sigma_X}\sqrt{p(1-p)}.$$

*Proof.* The correlation $\rho_{X,Y}$ is defined as:

$$\rho_{X,Y} = \frac{\mathrm{Cov}(X,Y)}{\sqrt{\mathrm{Var}(X)\mathrm{Var}(Y)}}.$$

We have $\mathrm{Var}(Y) = p(1-p)$. Using the law of total covariance:

$$\begin{aligned} \mathrm{Cov}(X,Y) &= \mathrm{E}(XY) - \mathrm{E}(X)\,\mathrm{E}(Y) \\ &= p\mu_1 - (p\mu_1 + (1-p)\mu_0)p \\ &= p(1-p)(\mu_1 - \mu_0). \end{aligned}$$

Since $\mathrm{Var}(Y) = p(1-p)$ the result follows. $\square$

**Corollary A.3** (Gaussian-Bernoulli Case). *Suppose $X$ and $Z$ are jointly Gaussian random variables with:*

$$X, Z \sim N(0,1), \quad \mathrm{Corr}(X, Z) = \rho,$$

*and define $Y = \mathbf{1}[Z > \alpha]$. Then the correlation between $X$ and $Y$ is:*

$$\rho_{X,Y} = \frac{\rho\,\phi(\alpha)}{\sqrt{\Phi(\alpha)(1-\Phi(\alpha))}},$$

*where $\phi(\alpha)$ and $\Phi(\alpha)$ are the PDF and CDF of the standard normal distribution, respectively. Further for $q = \Phi(\alpha)$, we have:*

$$\rho_{X,Y} \to 0 \quad as \quad q \to 1.$$

*Proof.* From the theorem above, we identify:

- $p = P(Z > \alpha) = 1 - \Phi(\alpha)$.

- Since $(X, Z)$ is jointly normal, $X|Z = z \sim N(\rho z, 1 - \rho^2)$, we have:

$$\mu_1 = E[X \mid Z > \alpha] = \rho E[Z \mid Z > \alpha] = \rho\frac{\phi(\alpha)}{1 - \Phi(\alpha)}.$$

- Similarly,

$$\mu_0 = E[X \mid Z \le \alpha] = \rho E[Z \mid Z \le \alpha] = -\rho\frac{\phi(\alpha)}{\Phi(\alpha)}.$$

Thus:

$$\mu_1 - \mu_0 = \rho\left(\frac{\phi(\alpha)}{1 - \Phi(\alpha)} + \frac{\phi(\alpha)}{\Phi(\alpha)}\right) = \frac{\rho\phi(\alpha)}{\Phi(\alpha)(1 - \Phi(\alpha))}.$$

Substituting these results into the theorem, we obtain:

$$\rho_{X,Y} = \frac{\rho\phi(\alpha)}{\Phi(\alpha)(1-\Phi(\alpha))}\sqrt{\Phi(\alpha)(1-\Phi(\alpha))}$$

$$= \frac{\rho\,\phi(\alpha)}{\sqrt{\Phi(\alpha)(1-\Phi(\alpha))}}.$$

Finally, note that $\rho(\alpha)$ is bounded hence $\rho_{X,Y} \to 0$ as $q = \Phi(\alpha) \to 0$. $\qquad\square$

One can also compute bounds for two correlated Bernoulli variables: Let $X \sim \text{Bernoulli}(p)$ and $Y \sim \text{Bernoulli}(q)$ with $0 < p, q < 1$. Their correlation coefficient is given by

$$\rho = \frac{P(X=1, Y=1) - pq}{\sqrt{p(1-p)q(1-q)}},$$

where the joint probability $P(X=1, Y=1)$ satisfies the Fréchet bounds:

$$\max\{0,\, p+q-1\} \le P(X=1, Y=1) \le \min\{p,\, q\}.$$

Thus, the extreme values of $\rho$ are:

$$\rho_{\max}(p, q) = \frac{\min\{p, q\} - pq}{\sqrt{p(1-p)q(1-q)}}$$

and

$$\rho_{\min}(p, q) = \frac{\max\{0,\, p+q-1\} - pq}{\sqrt{p(1-p)q(1-q)}}.$$

# B  Additional Theory

## B.1  Maximum–Absolute and Min–Max Normalization for Normally Distributed Data

In Theorem B.1, we show that the scaling factor in the max–abs method converges in distribution to a Gumbel distribution.

**Theorem B.1.** *Let $X_1, X_2, \ldots, X_n$ be a sample of normally distributed random variables, each with mean $\mu$ and standard deviation $\sigma$. Then*

$$\lim_{n\to\infty} \Pr\left(\max_{i\in[n]} |X_i| \le x\right) = G(x),$$

*where $G$ is the cumulative distribution function of a Gumbel distribution with parameters*

$$b_n = F_Y^{-1}(1-1/n) \quad and \quad a_n = \frac{1}{n f_Y(\mu_n)},$$

*where $f_Y$ and $F_Y^{-1}$ are the probability distribution function and quantile function, respectively, of a folded normal distribution with mean $\mu$ and standard deviation $\sigma$.*

The gist of Theorem B.1 is that the limiting distribution of $\max_{i\in[n]} |X_i|$ has expected value $b_n + \gamma a_n$, where $\gamma$ is the Euler-Mascheroni constant, which shows that the scaling factor depends on the sample size. In Figure 11(a), we observe empirically that the limiting distribution agrees well with the empirical distribution in expected value even for small values of $n$.

In Figure 11(b) we show the effect of increasing the number of observations, $n$, in a two-feature lasso model with max-abs normalization applied to both features. The coefficient corresponding to the Normally distributed feature shrinks as the number of observation $n$ increases. Since the expected value of the Gumbel distribution diverges with $n$, this means that there's always a large enough $n$ to force the coefficient in a lasso problem to zero with high probability.

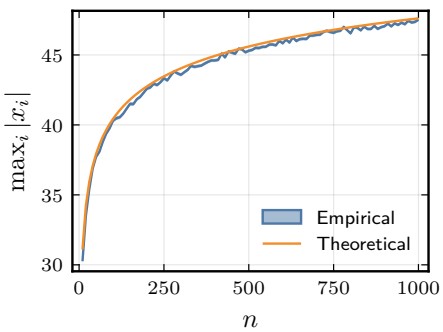

(a) Theoretical versus empirical distribution of the maximum absolute value of normally distributed random variables.

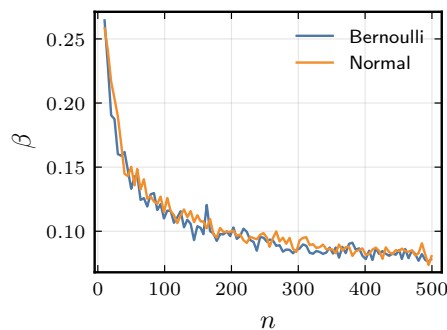

(b) Estimation of mixed features under maximum absolute value scaling

Figure 11: Effects of maximum absolute value scaling

For min–max normalization, the situation is similar and we omit the details here. The main point is that the scaling factor is strongly dependent on the sample size, which makes it unsuitable for normally distributed data in several situations, such as on-line learning (where sample size changes over time) or model validation with uneven data splits.

## B.2   Solution to the Elastic Net

Let $(\hat{\beta}_0^{(n)}, \hat{\boldsymbol{\beta}}^{(n)})$ be a solution to the problem in Equation (2). Expanding the function, we have

$$\frac{1}{2}\left(\boldsymbol{y}^\mathsf{T}\boldsymbol{y} - 2(\tilde{\boldsymbol{X}}\boldsymbol{\beta} + \beta_0)^\mathsf{T}\boldsymbol{y} + (\tilde{\boldsymbol{X}}\boldsymbol{\beta} + \beta_0)^\mathsf{T}(\tilde{\boldsymbol{X}}\boldsymbol{\beta} + \beta_0)\right) + \lambda_1\|\boldsymbol{\beta}\|_1 + \frac{\lambda_2}{2}\|\boldsymbol{\beta}\|_2^2.$$

Taking the subdifferential with respect to $\boldsymbol{\beta}$ and $\beta_0$, the KKT stationarity condition yields the following system of equations:

$$\begin{cases} \tilde{\boldsymbol{X}}^\mathsf{T}(\tilde{\boldsymbol{X}}\boldsymbol{\beta} + \beta_0 - \boldsymbol{y}) + \lambda_1 g + \lambda_2\boldsymbol{\beta} \ni \boldsymbol{0}, \\ n\beta_0 + (\tilde{\boldsymbol{X}}\boldsymbol{\beta})^\mathsf{T}\boldsymbol{1} - \boldsymbol{y}^\mathsf{T}\boldsymbol{1} = 0, \end{cases} \tag{11}$$

where $g$ is a subgradient of the $\ell_1$ norm that has elements $g_i$ such that

$$g_i \in \begin{cases} \{\text{sign } \beta_i\} & \text{if } \beta_i \neq 0, \\ [-1, 1] & \text{otherwise.} \end{cases}$$

### B.2.1   Orthogonal Features

If the features of the normalized design matrix are orthogonal, that is, $\tilde{\boldsymbol{X}}^\mathsf{T}\tilde{\boldsymbol{X}} = \text{diag}\left(\tilde{\boldsymbol{x}}_1^\mathsf{T}\tilde{\boldsymbol{x}}_1, \dots, \tilde{\boldsymbol{x}}_p^\mathsf{T}\tilde{\boldsymbol{x}}_p\right)$, then Equation (11) can be decomposed into a set of $p + 1$ conditions:

$$\begin{cases} \tilde{\boldsymbol{x}}_j^\mathsf{T}(\tilde{\boldsymbol{x}}_j\beta_j + \boldsymbol{1}\beta_0 - \boldsymbol{y}) + \lambda_2\beta_j + \lambda_1 g \ni 0, \quad j \in [p], \\ n\beta_0 + (\tilde{\boldsymbol{X}}\boldsymbol{\beta})^\mathsf{T}\boldsymbol{1} - \boldsymbol{y}^\mathsf{T}\boldsymbol{1} = 0. \end{cases}$$

The inclusion of the intercept ensures that the locations (means) of the features do not affect the solution (except for the intercept itself). We will therefore from now on assume that the features are mean-centered so that $c_j = \bar{\boldsymbol{x}}_j$ for all $j$ and therefore $\tilde{\boldsymbol{x}}_j^\mathsf{T}\boldsymbol{1} = 0$. A solution to the system of equations is then given by the following set of equations (Donoho & Johnstone, 1994):

$$\hat{\beta}_j^{(n)} = \frac{\text{S}_{\lambda_1}\left(\tilde{\boldsymbol{x}}_j^\mathsf{T}\boldsymbol{y}\right)}{\tilde{\boldsymbol{x}}_j^\mathsf{T}\tilde{\boldsymbol{x}}_j + \lambda_2}, \qquad \hat{\beta}_0^{(n)} = \frac{\boldsymbol{y}^\mathsf{T}\boldsymbol{1}}{n},$$

where $\text{S}_\lambda(z) = \text{sign}(z)\max(|z| - \lambda, 0)$ is the soft-thresholding operator.

## B.3 Bias and Variance of the Elastic Net Estimator

Here, we derive the results in Section 3 in more detail. Let

$$Z_j = \tilde{\boldsymbol{x}}_j^\mathsf{T} \boldsymbol{y} = \tilde{\boldsymbol{x}}_j^\mathsf{T} (\boldsymbol{X}\boldsymbol{\beta}^* + \boldsymbol{\varepsilon}) = \tilde{\boldsymbol{x}}_j^\mathsf{T} (\boldsymbol{x}_j \beta_j^* + \boldsymbol{\varepsilon}) \qquad \text{and} \qquad d_j = s_j (\tilde{\boldsymbol{x}}_j^\mathsf{T} \tilde{\boldsymbol{x}}_j + \lambda_2)$$

so that $\hat{\beta}_j = \mathrm{S}_{\lambda_1}(Z_j)/d_j$. Since $d_j$ is fixed under our assumptions, we focus on $\mathrm{S}_{\lambda_1}(Z_j)$. First observe that since $c_j = \bar{\boldsymbol{x}}_j$,

$$\tilde{\boldsymbol{x}}_j^\mathsf{T} \tilde{\boldsymbol{x}}_j = \frac{1}{s_j^2}(\boldsymbol{x}_j - c_j)^\mathsf{T}(\boldsymbol{x}_j - c_j) = \frac{\boldsymbol{x}_j^\mathsf{T}\boldsymbol{x}_j - nc_j^2}{s_j^2} = \frac{n\nu_j}{s_j^2},$$

$$\tilde{\boldsymbol{x}}_j^\mathsf{T} \boldsymbol{x}_j = \frac{1}{s_j}(\boldsymbol{x}_j^\mathsf{T}\boldsymbol{x}_j - \boldsymbol{x}_j^\mathsf{T} \mathbf{1}c_j) = \frac{n\nu_j}{s_j},$$

where $\nu_j$ is the uncorrected sample variance of $\boldsymbol{x}_j$. This means that

$$Z_j = \tilde{\boldsymbol{x}}_j^\mathsf{T}(\boldsymbol{x}_j \beta_j^* + \boldsymbol{\varepsilon}) = \frac{\beta_j^* n\nu_j - \boldsymbol{x}_j^\mathsf{T}\boldsymbol{\varepsilon} + c_j \mathbf{1}^\mathsf{T}\boldsymbol{\varepsilon}}{s_j} \qquad \text{and} \qquad d_j = s_j \left( \frac{n\nu_j}{s_j^2} + \lambda_2 \right). \tag{12}$$

For the expected value and variance of $Z_j$ we then have

$$\mathrm{E}\, Z_j = \mu_j = \mathrm{E}\left( \tilde{\boldsymbol{x}}_j^\mathsf{T}(\boldsymbol{x}_j \beta_j^* + \boldsymbol{\varepsilon}) \right) = \tilde{\boldsymbol{x}}_j^\mathsf{T}\boldsymbol{x}_j \beta_j^* = \frac{\beta_j^* n\nu_j}{s_j},$$

$$\mathrm{Var}\, Z_j = \sigma_j^2 = \mathrm{Var}\left( \tilde{\boldsymbol{x}}_j^\mathsf{T}\boldsymbol{\varepsilon} \right) = \tilde{\boldsymbol{x}}_j^\mathsf{T}\tilde{\boldsymbol{x}}_j \sigma_\varepsilon^2 = \frac{n\nu_j \sigma_\varepsilon^2}{s_j^2}.$$

The expected value of the soft-thresholding estimator is

$$\mathrm{E}\, \mathrm{S}_\lambda(Z_j) = \int_{-\infty}^\infty \mathrm{S}_\lambda(z) f_{Z_j}(z)\, \mathrm{d}z = \int_{-\infty}^{-\lambda} (z + \lambda) f_{Z_j}(z)\, \mathrm{d}z + \int_\lambda^\infty (z - \lambda) f_{Z_j}(z)\, \mathrm{d}z.$$

And then the bias of $\hat{\beta}_j$ with respect to the true coefficient $\beta_j^*$ is

$$\mathrm{E}\, \hat{\beta}_j - \beta_j^* = \frac{1}{d_j} \mathrm{E}\, \mathrm{S}_\lambda(Z_j) - \beta_j^*.$$

Finally, we note that the variance of the soft-thresholding estimator is

$$\mathrm{Var}\, \mathrm{S}_\lambda(Z_j) = \int_{-\infty}^{-\lambda} (z + \lambda)^2 f_{Z_j}(z)\, \mathrm{d}z + \int_\lambda^\infty (z - \lambda)^2 f_{Z_j}(z)\, \mathrm{d}z - \left( \mathrm{E}\, \mathrm{S}_\lambda(Z_j) \right)^2 \tag{13}$$

and that the variance of the elastic net estimator is therefore

$$\mathrm{Var}\, \hat{\beta}_j = \frac{1}{d_j^2} \mathrm{Var}\, \mathrm{S}_\lambda(Z_j).$$

### B.3.1 Normally Distributed Noise

We now assume that $\boldsymbol{\varepsilon}$ is normally distributed. Then

$$Z_j \sim \mathrm{Normal}\left( \mu_j = \tilde{\boldsymbol{x}}_j^\mathsf{T}\boldsymbol{x}_j \beta_j^*, \sigma_j^2 = \tilde{\boldsymbol{x}}_j^\mathsf{T}\tilde{\boldsymbol{x}}_j \sigma_\varepsilon^2 \right).$$

Let $\theta_j = -\mu_j - \lambda_1$ and $\gamma_j = \mu_j - \lambda_1$. Then the expected value of soft-thresholding of $Z_j$ is

$$\mathrm{E}\, \mathrm{S}_{\lambda_1}(Z_j) = \int_{-\infty}^{\frac{\theta_j}{\sigma_j}} (\sigma_j u - \theta_j)\, \phi(u)\, \mathrm{d}u + \int_{-\frac{\gamma_j}{\sigma_j}}^\infty (\sigma_j u + \gamma_j)\, \phi(u)\, \mathrm{d}u$$

$$= -\theta_j\, \Phi\left( \frac{\theta_j}{\sigma_j} \right) - \sigma_j\, \phi\left( \frac{\theta_j}{\sigma_j} \right) + \gamma_j\, \Phi\left( \frac{\gamma_j}{\sigma_j} \right) + \sigma_j\, \phi\left( \frac{\gamma_j}{\sigma_j} \right)$$

where $\phi(u)$ and $\Phi(u)$ are the probability density and cumulative distribution functions of the standard normal distribution, respectively. Computing Equation (13) gives us

$$
\begin{aligned}
\operatorname{Var} S_\lambda(Z_j) = {}& \frac{\sigma_j^2}{2}\left(\operatorname{erf}\left(\frac{\theta_j}{\sigma_j\sqrt{2}}\right) \quad \frac{\theta_j}{\sigma_j}\sqrt{\frac{2}{\pi}}\exp\left(-\frac{\theta_j^2}{2\sigma_j^2}\right) + 1\right) \\
& + 2\theta_j\sigma_j\,\phi\left(\frac{\theta_j}{\sigma_j}\right) + \theta_j^2\,\Phi\left(\frac{\theta_j}{\sigma_j}\right) \\
& + \frac{\sigma_j^2}{2}\left(\operatorname{erf}\left(\frac{\gamma_j}{\sigma_j\sqrt{2}}\right) - \frac{\gamma_j}{\sigma_j}\sqrt{\frac{2}{\pi}}\exp\left(-\frac{\gamma_j^2}{2\sigma_j^2}\right) + 1\right) \\
& + 2\gamma_j\sigma_j\,\phi\left(\frac{\gamma_j}{\sigma_j}\right) + \gamma_j^2\,\Phi\left(\frac{\gamma_j}{\sigma_j}\right) \\
& - \left(\operatorname{E} S_{\lambda_1}(Z_j)\right)^2.
\end{aligned}
$$

### B.4 Derivation of Estimate in the Noiseless Case

If we assume $s_j = (q_j - q_j^2)^\delta$, then Equation (7) becomes

$$
\hat{\beta}_j = \frac{S_{\lambda_1}\left(\beta_j^* n(q_j - q_j^2)^{1-\delta}\right)}{n(q_j - q_j^2)^{1-\delta} + (q_j - q_j^2)^\delta \lambda_2}.
$$

If we are in the lasso case, then $\lambda_2 = 0$ and

$$
\hat{\beta}_j = \frac{S_{\lambda_1}\left(\beta_j^* n(q_j - q_j^2)^{1-\delta}\right)}{n(q_j - q_j^2)^{1-\delta}},
$$

so we need to choose $\delta = 1$, and hence $s_j = q_j - q_j^2$, to get rid of the dependency on $q_j$. For the ridge case, we instead have $\lambda_1 = 0$ and hence

$$
\hat{\beta}_j = \frac{\beta_j^* n(q_j - q_j^2)^{1-\delta}}{n(q_j - q_j^2)^{1-\delta} + (q_j - q_j^2)^\delta \lambda_2} = \frac{\beta_j^* n}{n + (q_j - q_j^2)^{2\delta-1}\lambda_2},
$$

which shows that we need to choose $\delta = 1/2$ and hence $s_j = \sqrt{q_j - q_j^2}$, to get rid of the dependency on $q_j$ in this case.

Note that if $\lambda_1, \lambda_2 > 0$ (the elastic net case), then there is no choice of $\delta$ that will make the estimator independent of $q_j$.

### B.5 Bias and Variance for Ridge Regression

**Corollary B.2** (Variance in Ridge Regression)**.** *Assume the conditions of Theorem 3.1 hold, except that* $\lambda_1 = 0$. *Then*

$$
\lim_{q_j \to 1^-} \operatorname{Var} \hat{\beta}_j = \begin{cases} 0 & \text{if } 0 \le \delta < 1/4, \\ \frac{\sigma_\varepsilon^2 n}{\lambda_2^2} & \text{if } \delta = 1/4, \\ \infty & \text{if } \delta > 1/4. \end{cases}
$$

### B.6 Centering and Interaction Features

The main motivation for centering is that it removes correlation between the main features and the interaction, which would otherwise affect the estimates due to the regularization. Centering normal features is also important because it ensures that their means do not factor into the estimation of their effects, which is otherwise the case since the variance of $\boldsymbol{x}_3$ would then be $q_1(\sigma^2 + \mu^2(1 - q_1))$ in the case when $\boldsymbol{x}_1$ is centered and $(q_1 - q_1^2)(\sigma^2 + \mu^2)$ otherwise. Centering binary features is also important because the variance of the interaction term is otherwise $q_1\sigma^2$ (provided $\boldsymbol{x}_2$ is centered), which would mean that the encoding of values of the binary feature (e.g. $\{0, 1\}$ versus $\{-1, 1\}$) would affect the interaction term.

### B.7 Extended Results on Bias and Variance for Ridge, Lasso, and Elastic Net Regression

In Figure 12, we show bias, variance, and mean-squared error for the weighted elastic net. We see that the behavior of bias as $q_j \to 1^-$ depends on noise level and that there is a bias–variance trade-off with respect to $\omega$. As in Section 3.2, we modify the weighting factor to have comparability under $\kappa = 2$.

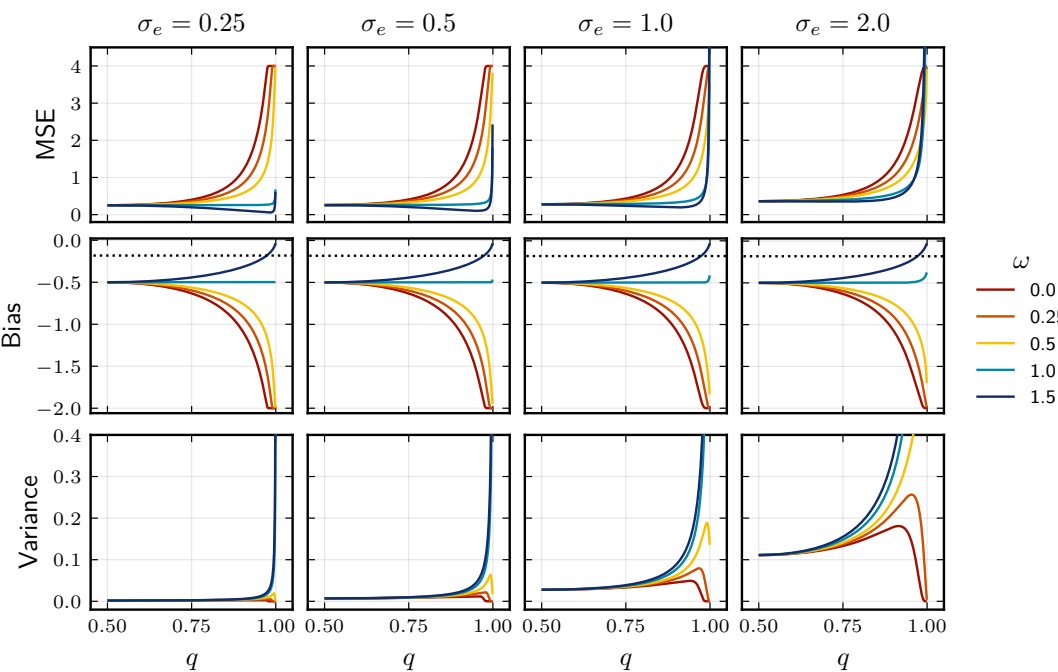

Figure 12: Bias, variance, and mean-squared error in the case of the one-dimensional weighted elastic net. The measures are shown for different noise levels ($\sigma_\varepsilon$), class balances ($q_j$), and values of ($\omega$), which controls the weights that are set to $u_j = v_j = 2 \times 4^{\omega-1}(q - q^2)^\omega$ in order for the results to be comparable across different values of $\omega$. The dotted lines represent the asymptotic bias of the estimator in the case of $\omega = 1$. In the case of $\omega > 1$, the limit of the bias is zero.

## C  Proofs

## D  Proof of Theorem 3.1

To avoid excessive notation, we allow ourselves to abuse notation and will drop the subscript $j$ everywhere in this proof, allowing $\beta^*$, $s$, and so on to respectively denote $\beta^*$, $s_j$ et cetera.

Since $s = (q - q^2)^\delta$, we have

$$\mu = \beta^* n(q - q^2)^{1-\delta}, \qquad \frac{\theta}{\sigma} = -a\sqrt{q - q^2} - b(q - q^2)^{\delta-1/2},$$

$$\sigma = \sigma_\varepsilon \sqrt{n}(q - q^2)^{1/2-\delta}, \qquad \frac{\gamma}{\sigma} = a\sqrt{q - q^2} - b(q - q^2)^{\delta-1/2},$$

$$d = n(q - q^2)^{1-\delta} + \lambda_2(q - q^2)^\delta, \qquad \frac{\theta}{d} = \frac{-\beta^* n - \lambda_1(q - q^2)^{\delta-1}}{n + \lambda_2(q - q^2)^{2\delta-1}},$$

$$\theta = -\beta^* n(q - q^2)^{1-\delta} - \lambda_1, \qquad \frac{\gamma}{d} = \frac{\beta^* n - \lambda_1(q - q^2)^{\delta-1}}{n + \lambda_2(q - q^2)^{2\delta-1}},$$

$$\gamma = \beta^* n(q - q^2)^{1-\delta} - \lambda_1,$$

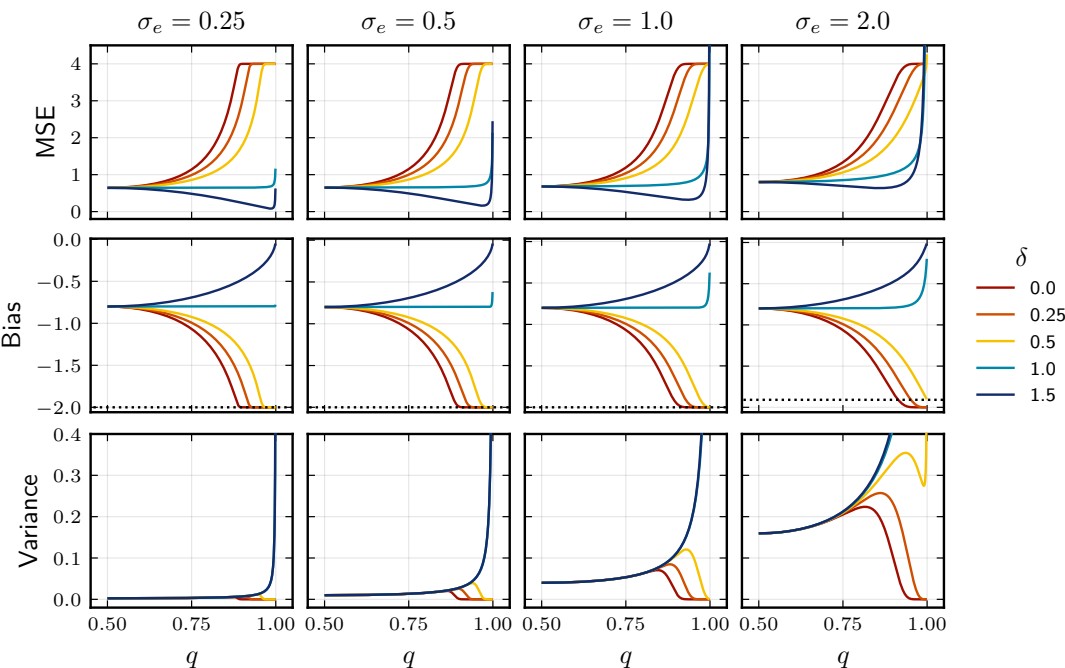

Figure 13: Bias, variance, and mean-squared error for a one-dimensional lasso problem, parameterized by noise level $(\sigma_\varepsilon)$, class balance $(q)$, and scaling $(\delta)$. Dotted lines represent asymptotic bias of the lasso estimator in the case when $\delta = 1/2$.

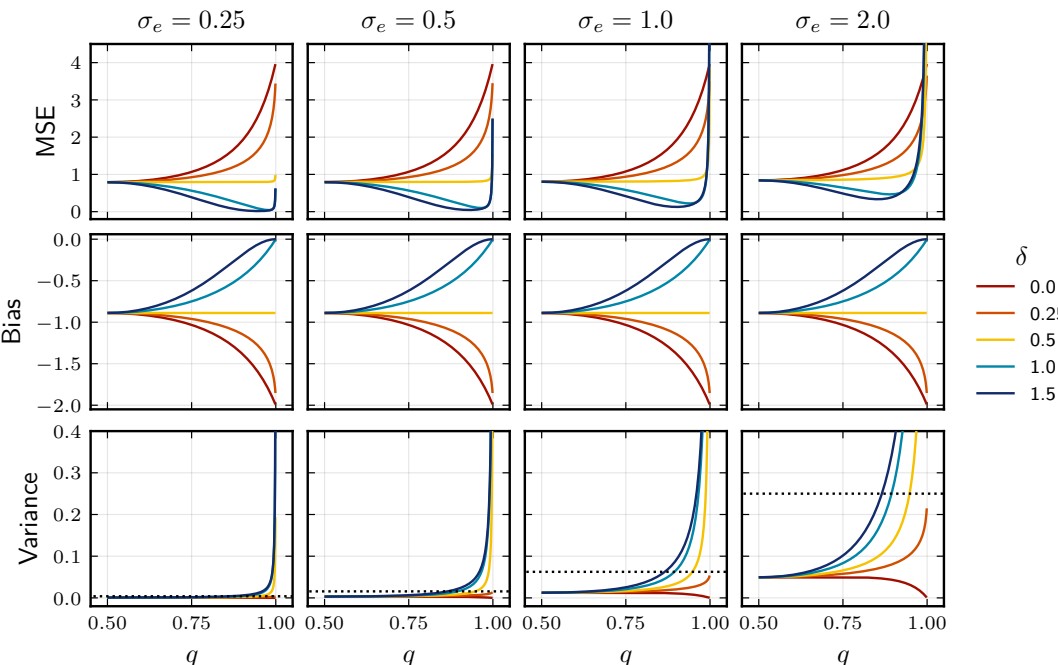

Figure 14: Bias, variance, and mean-squared error for one-dimensional ridge regression, parameterized by noise level $(\sigma_\varepsilon)$, class balance $(q)$, and scaling $(\delta)$. Dotted lines represent asymptotic bias of the ridge estimator in the case of $\delta = 1/4$.

with

$$a = \frac{\beta^* \sqrt{n}}{\sigma_\varepsilon} \qquad \text{and} \qquad b = \frac{\lambda_1}{\sigma_\varepsilon \sqrt{n}}.$$

We are interested in

$$\lim_{q \to 1^-} \mathrm{E}\,\hat{\beta} = \lim_{q \to 1^-} \frac{1}{d} \left( -\theta\,\Phi\left(\frac{\theta}{\sigma}\right) - \sigma\,\phi\left(\frac{\theta}{\sigma}\right) + \gamma\,\Phi\left(\frac{\gamma}{\sigma}\right) + \sigma\,\phi\left(\frac{\gamma}{\sigma}\right) \right). \tag{14}$$

Before we proceed, note the following limits, which we will make repeated use of throughout the proof.

$$\lim_{q \to 1^-} \frac{\theta}{\sigma} = \lim_{q \to 1^-} \frac{\gamma}{\sigma} = \begin{cases} -\infty & \text{if } 0 \le \delta < \frac{1}{2}, \\ -b & \text{if } \delta = \frac{1}{2}, \\ 0 & \text{if } \delta > \frac{1}{2}, \end{cases} \tag{15}$$

Starting with the terms involving $\Phi$ inside the limit in Equation (14), for now assuming that they are well-defined and that the limits of the remaining terms also exist seperately, we have

$$\lim_{q \to 1^-} \left( -\frac{\theta}{d}\,\Phi\left(\frac{\theta}{\sigma}\right) + \frac{\gamma}{d}\,\Phi\left(\frac{\gamma}{\sigma}\right) \right) = \lim_{q \to 1^-} \left( \left( \frac{\beta^* n}{n + \lambda_2(q - q^2)^{2\delta - 1}} + \frac{\lambda_1}{n(q - q^2)^{1-\delta} + \lambda_2(q - q^2)^\delta} \right) \Phi\left(\frac{\theta}{\sigma}\right) \right.$$

$$\left. + \left( \frac{\beta^* n}{n + \lambda_2(q - q^2)^{2\delta - 1}} - \frac{\lambda_1}{n(q - q^2)^{1-\delta} + \lambda_2(q - q^2)^\delta} \right) \Phi\left(\frac{\gamma}{\sigma}\right) \right)$$

$$= \lim_{q \to 1^-} \frac{\beta^* n}{n + \lambda_2(q - q^2)^{2\delta - 1}} \left( \Phi\left(\frac{\theta}{\sigma}\right) + \Phi\left(\frac{\gamma}{\sigma}\right) \right)$$

$$+ \lim_{q \to 1^-} \frac{\lambda_1}{n(q - q^2)^{1-\delta} + \lambda_2(q - q^2)^\delta} \left( \Phi\left(\frac{\theta}{\sigma}\right) - \Phi\left(\frac{\gamma}{\sigma}\right) \right). \tag{16}$$

Considering the first term in Equation (16), we see that

$$\lim_{q \to 1^-} \frac{\beta^* n}{n + \lambda_2(q - q^2)^{2\delta - 1}} \left( \Phi\left(\frac{\theta}{\sigma}\right) + \Phi\left(\frac{\gamma}{\sigma}\right) \right) = \begin{cases} 0 & \text{if } 0 \le \delta < 1/2, \\ \frac{2n\beta^*}{n + \lambda_2}\,\Phi(-b) & \text{if } \delta = 1/2, \\ \beta^* & \text{if } \delta > 1/2. \end{cases}$$

For the second term in Equation (16), we start by observing that if $\delta = 1$, then $(q - q^2)^{\delta - 1} = 1$, and if $\delta > 1$, then $\lim_{q \to 1^-} (q - q^2)^{\delta - 1} = 0$. Moreover, the arguments of $\Phi$ approach 0 in the limit for $\delta \ge 1$, which means that the entire term vanishes in both cases ($\delta \ge 1$).

For $0 \le \delta < 1$, the limit is indeterminite of the form $\infty \times 0$. We define

$$f(q) = \Phi\left(\frac{\theta}{\sigma}\right) - \Phi\left(\frac{\gamma}{\sigma}\right) \qquad \text{and} \qquad g(q) = n(q - q^2)^{1-\delta} + \lambda_2(q - q^2)^\delta,$$

such that we can express the limit as $\lim_{q \to 1^-} f(q)/g(q)$. The corresponding derivatives are

$$f'(q) = \left( -\frac{a}{2}(1 - 2q)(q - q^2)^{-1/2} - b(\delta - 1/2)(1 - 2q)(q - q^2)^{\delta - 3/2} \right) \phi\left(\frac{\theta}{\sigma}\right)$$

$$- \left( \frac{a}{2}(1 - 2q)(q - q^2)^{-1/2} - b(\delta - 1/2)(1 - 2q)(q - q^2)^{\delta - 3/2} \right) \phi\left(\frac{\gamma}{\sigma}\right),$$

$$g'(q) = n(1 - \delta)(1 - 2q)(q - q^2)^{-\delta} + \lambda_2\delta(1 - 2q)(q - q^2)^{\delta - 1}$$

Note that $f(q)$ and $g(q)$ are both differentiable and $g'(q) \ne 0$ everywhere in the interval $(1/2, 1)$. Now note that we have

$$\frac{f'(q)}{g'(q)} = \frac{1}{n(1 - \delta)(q - q^2)^{1/2 - \delta} + \lambda_2\delta(1 - 2q)(q - q^2)^{\delta - 1/2}}$$

$$\times \left( -\left( \frac{a}{2} + b(\delta - 1/2)(q - q^2)^{\delta - 1} \right) \phi\left(\frac{\theta}{\sigma}\right) - \left( \frac{a}{2} - b(\delta - 1/2)(q - q^2)^{\delta - 1} \right) \phi\left(\frac{\gamma}{\sigma}\right) \right). \tag{17}$$

For $0 \leq \delta < 1/2$, $\lim_{q \to 1^-} f'(q)/g'(q) = 0$ since the exponential terms of $\phi$ in Equation (17) dominate in the limit.

For $\delta = 1/2$, we have

$$\lim_{q \to 1^-} \frac{f'(q)}{g'(q)} = -\frac{a}{n + \lambda_2} \lim_{q \to 1^-} \left( \phi\left(\frac{\theta}{\sigma}\right) + \phi\left(\frac{\gamma}{\sigma}\right) \right) = -\frac{2a\,\phi(-b)}{n + \lambda_2}$$

so that we can use L'Hôpital's rule to show that the second term in Equation (16) becomes

$$-\frac{2\beta^* \lambda_1 \sqrt{n}}{\sigma_\varepsilon(n + \lambda_2)} \phi\left(\frac{-\lambda_1}{\sigma_\varepsilon \sqrt{n}}\right). \tag{18}$$

For $\delta > 1/2$, we have

$$\lim_{q \to 1^-} \frac{f'(q)}{g'(q)} = \lim_{q \to 1^-} \frac{-\frac{a}{2}\left(\phi\left(\frac{\theta}{\sigma}\right) + \phi\left(\frac{\gamma}{\sigma}\right)\right)}{n(1-\delta)(q-q^2)^{1/2-\delta} + \lambda_2\delta(1-2q)(q-q^2)^{\delta-1/2}}$$

$$+ \lim_{q \to 1^-} \frac{b(\delta - 1/2)\left(\phi\left(\frac{\gamma}{\sigma}\right) - \phi\left(\frac{\theta}{\sigma}\right)\right)}{n(1-\delta)(q-q^2)^{3/2-2\delta} + \lambda_2\delta(1-2q)(q-q^2)^{1/2}}$$

$$= 0 + \lim_{q \to 1^-} \frac{b(\delta - 1/2)e^{-\frac{1}{2}\left(a^2(q-q^2)+b^2(q-q^2)^{2\delta-1}\right)}\left(e^{-ab(q-q^2)^\delta} - e^{ab(q-q^2)^\delta}\right)}{\sqrt{2\pi}\left(n(1-\delta)(q-q^2)^{3/2-2\delta} + \lambda_2\delta(1-2q)(q-q^2)^{1/2}\right)}$$

$$= 0$$

since the exponential term in the numerator dominates.

Now we proceed to consider the terms involving $\phi$ in Equation (14). We have

$$\lim_{q \to 1^-} \frac{\sigma}{d}\left(\phi\left(\frac{\gamma}{\sigma}\right) - \phi\left(\frac{\theta}{\sigma}\right)\right) = \sigma_\varepsilon \sqrt{n} \lim_{q \to 1^-} \frac{\phi\left(\frac{\gamma}{\sigma}\right) - \phi\left(\frac{\theta}{\sigma}\right)}{n(q-q^2)^{1/2} + \lambda_2(q-q^2)^{2\delta-1/2}} \tag{19}$$

For $0 \leq \delta < 1/2$, we observe that the exponential terms in $\phi$ dominate in the limit, and so we can distribute the limit and consider the limits of the respective terms individually, which both vanish.

For $\delta \geq 1/2$, the limit in Equation (19) has an indeterminate form of the type $\frac{0}{0}$. Define

$$u(q) = \phi\left(\frac{\gamma}{\sigma}\right) - \phi\left(\frac{\theta}{\sigma}\right) \qquad \text{and} \qquad v(q) = n(q-q^2)^{1/2} + \lambda_2(q-q^2)^{2\delta-1/2}$$

which are both differentiable in the interval $(1/2, 1)$ and $v'(q) \neq 0$ everywhere in this interval. The derivatives are

$$u'(q) = -\phi\left(\frac{\gamma}{\sigma}\right)\frac{\gamma}{\sigma}\left(\frac{1}{2}\left(a(1-2q)(q-q^2)^{-1/2}\right) - b(\delta-1/2)(1-2q)(q-q^2)^{\delta-3/2}\right)$$

$$+ \phi\left(\frac{\theta}{\sigma}\right)\frac{\theta}{\sigma}\left(\frac{1}{2}\left(a(1-2q)(q-q^2)^{-1/2}\right) + b(\delta-1/2)(1-2q)(q-q^2)^{\delta-3/2}\right),$$

$$v'(q) = \frac{n}{2}(1-2q)(q-q^2)^{-1/2} + \lambda_2(2\delta-1/2)(1-2q)(q-q^2)^{2\delta-3/2}.$$

And so

$$\frac{u'(q)}{v'(q)} = \frac{1}{n + \lambda_2(4\delta-1)(q-q^2)^{2\delta-1}}\Bigg( \left(a - b(2\delta-1)(q-q^2)^{\delta-1}\right)\phi\left(\frac{\gamma}{\sigma}\right)\frac{\gamma}{\sigma}$$

$$+ \left(a + b(2\delta-1)(q-q^2)^{\delta-1}\right)\phi\left(\frac{\theta}{\sigma}\right)\frac{\theta}{\sigma}\Bigg). \tag{20}$$

Taking the limit, rearranging, and assuming that the limits of the separate terms exist, we obtain

$$
\begin{aligned}
\lim_{q\to 1^-}\frac{u'(q)}{v'(q)} &= a\lim_{q\to 1^-}\frac{1}{n+\lambda_2(4\delta-1)(q-q^2)^{2\delta-1}}\left(\phi\left(\frac{\gamma}{\sigma}\right)\frac{\gamma}{\sigma}-\phi\left(\frac{\theta}{\sigma}\right)\frac{\theta}{\sigma}\right)\\
&+ b(2\delta-1)\lim_{q\to 1^-}\frac{1}{n+\lambda_2(4\delta-1)(q-q^2)^{2\delta-1}}\left(\phi\left(\frac{\gamma}{\sigma}\right)\left(a(q-q^2)^{\delta-1/2}-b(q-q^2)^{2\delta-3/2}\right)\right.\\
&\left. \qquad\qquad -\phi\left(\frac{\theta}{\sigma}\right)\left(-a(q-q^2)^{\delta-1/2}-b(q-q^2)^{2\delta-3/2}\right)\right). \quad (21)
\end{aligned}
$$

For $\delta=1/2$, we have

$$
\lim_{q\to 1^-}\frac{u'(q)}{v'(q)}=-\frac{a}{n+\lambda_2}\left(-b\,\phi(-b)-b\,\phi(-b)\right)+0=2ab\,\phi(-b)=\frac{2\beta^*\lambda_1}{\sigma_\varepsilon^2(n+\lambda_2)}\,\phi\left(\frac{-\lambda_1}{\sigma_\varepsilon\sqrt{n}}\right).
$$

Using L'Hôpital's rule, Equation (19) must consequently be

$$
\frac{2\beta^*\lambda_1\sqrt{n}}{\sigma_\varepsilon(n+\lambda_2)}\,\phi\left(\frac{-\lambda_1}{\sigma_\varepsilon\sqrt{n}}\right),
$$

which cancels with Equation (18).

For $\delta>1/2$, we first observe that the first term in Equation (21) tends to zero due to Equation (15) and the properties of the standard normal distribution. For the second term, we note that this is essentially of the same form as Equation (17) and that the limit is therefore 0 here.

### D.1  Proof of Theorem 3.2

The variance of the elastic net estimator is given by

$$
\begin{aligned}
\operatorname{Var}\hat{\beta}_j = \frac{1}{d^2}\Bigg(&\frac{\sigma^2}{2}\left(2+\operatorname{erf}\left(\frac{\theta}{\sigma\sqrt{2}}\right)-\frac{\theta}{\sigma}\sqrt{\frac{2}{\pi}}\exp\left(-\frac{\theta^2}{2\sigma^2}\right)+\operatorname{erf}\left(\frac{\gamma}{\sigma\sqrt{2}}\right)-\frac{\gamma}{\sigma}\sqrt{\frac{2}{\pi}}\exp\left(-\frac{\gamma^2}{2\gamma^2}\right)\right)\\
&+2\theta\sigma\,\phi\left(\frac{\theta}{\sigma}\right)+\theta^2\,\Phi\left(\frac{\theta}{\sigma}\right)+2\gamma\sigma\,\phi\left(\frac{\gamma}{\sigma}\right)+\gamma^2\,\Phi\left(\frac{\gamma}{\sigma}\right)\Bigg)-\left(\frac{1}{d}\operatorname{E}\hat{\beta}_j\right)^2. \quad (22)
\end{aligned}
$$

We start by noting the following identities:

$$
\begin{aligned}
\theta^2 &= (\beta^*n)^2\,(q-q^2)^{2-2\delta}+\lambda_1^2+2\lambda_1\beta^*n(q-q^2)^{1-\delta},\\
d^2 &= n^2(q-q^2)^{2-2\delta}+2n\lambda_2(q-q^2)+\lambda_2^2(q-q^2)^{2\delta},\\
\theta\sigma &= -\sigma_\varepsilon\left(\beta^*n^{3/2}(q-q^2)^{3/2-2\delta}+\sqrt{n}\lambda_1(q-q^2)^{1/2-\delta}\right),\\
\frac{\theta^2}{\sigma^2} &= a^2(q-q^2)+b^2(q-q^2)^{2\delta-1}+2ab(q-q^2)^\delta,\\
\frac{\sigma}{d} &= \frac{\sigma_\varepsilon\sqrt{n}}{n(q-q^2)^{\frac{1}{2}}+\lambda_2(q-q^2)^{2\delta-1/2}}.
\end{aligned}
$$

Expansions involving $\gamma$ instead of $\theta$ have identical expansions up to sign changes of the individual terms. Also recall the definitions provided in the proof of Theorem 3.1.

Starting with the case when $0 \le \delta < 1/2$, we write the limit of Equation (22) as

$$\lim_{q \to 1^-} \operatorname{Var} \hat{\beta}_j$$

$$= \sigma_\varepsilon^2 n \lim_{q \to 1^-} \frac{1}{\left(n(q-q^2)^{1/2} + \lambda_2(q-q^2)^{2\delta-1/2}\right)^2} \left(1 + \operatorname{erf}\left(\frac{\theta}{\sigma\sqrt{2}}\right) - \frac{\theta}{\sigma}\sqrt{\frac{2}{\pi}}\exp\left(-\frac{\theta^2}{2\sigma^2}\right)\right)$$

$$+ \sigma_\varepsilon^2 n \lim_{q \to 1^-} \frac{1}{\left(n(q-q^2)^{1/2} + \lambda_2(q-q^2)^{2\delta-1/2}\right)^2} \left(1 + \operatorname{erf}\left(\frac{\gamma}{\sigma\sqrt{2}}\right) - \frac{\gamma}{\sigma}\sqrt{\frac{2}{\pi}}\exp\left(-\frac{\gamma^2}{2\sigma^2}\right)\right)$$

$$+ \lim_{q \to 1^-} \frac{2\theta\sigma}{d^2} \phi\left(\frac{\theta}{\sigma}\right) + \lim_{q \to 1^-} \frac{\theta^2}{d^2} \Phi\left(\frac{\theta}{\sigma}\right) + \lim_{q \to 1^-} \frac{2\gamma}{d^2}\sigma\phi\left(\frac{\gamma}{\sigma}\right) + \lim_{q \to 1^-} \frac{\gamma^2}{d^2} \Phi\left(\frac{\gamma}{\sigma}\right)$$

$$- \left(\lim_{q \to 1^-} \frac{1}{d} \operatorname{E}\hat{\beta}_j\right)^2,$$

assuming, for now, that all limits exist. Next, let

$$f_1(q) = 1 + \operatorname{erf}\left(\frac{\theta}{\sigma\sqrt{2}}\right) - \frac{\theta}{\sigma}\sqrt{\frac{2}{\pi}}\exp\left(-\frac{\theta^2}{2\sigma^2}\right),$$

$$f_2(q) = 1 + \operatorname{erf}\left(\frac{\gamma}{\sigma\sqrt{2}}\right) - \frac{\gamma}{\sigma}\sqrt{\frac{2}{\pi}}\exp\left(-\frac{\gamma^2}{2\sigma^2}\right),$$

$$g(q) = \left(n^2(q-q^2) + 2n\lambda_2(q-q^2)^{2\delta} + \lambda_2^2(q-q^2)^{4\delta-1}\right)^2.$$

And

$$f_1'(q) = \frac{\theta^2}{\sigma^2}\sqrt{\frac{2}{\pi}}\exp\left(-\frac{\theta^2}{2\sigma^2}\right),$$

$$f_2'(q) = \frac{\gamma^2}{\sigma^2}\sqrt{\frac{2}{\pi}}\exp\left(-\frac{\gamma^2}{2\sigma^2}\right),$$

$$g'(q) = (1 - 2q)\left((q-q^2)^{-1} + 4n\delta\lambda_2(q-q^2)^{2\delta-1} + \lambda_2^2(4\delta-1)(q-q^2)^{4\delta-2}\right).$$

$f_1$, $f_1$ and $g$ are differentiable in $(1/2, 1)$ and $g'(q) \ne 0$ everywhere in this interval. $f_1/g$ and $f_2/g$ are indeterminate of the form $0/0$. And we see that

$$\lim_{q \to 1^-} \frac{f_1'(q)}{g'(q)} = \lim_{q \to 1^-} \frac{f_2'(q)}{g'(q)} = 0$$

due to the dominance of the exponential terms as $\theta/\sigma$ and $\gamma/\sigma$ both tend to $-\infty$. Thus $f_1/g$ and $f_2/g$ also tend to 0 by L'Hôpital's rule. Similar reasoning shows that

$$\lim_{q \to 1^-} \frac{2\theta\sigma}{d^2} \phi\left(\frac{\theta}{\sigma}\right) = \lim_{q \to 1^-} \frac{\theta^2}{d^2} \Phi\left(\frac{\theta}{\sigma}\right) = 0.$$

The same result applies to the respective terms involving $\gamma$. And since we in Theorem 3.1 showed that $\lim_{q \to 1^-} \frac{1}{d} \operatorname{E}\hat{\beta}_j = 0$, the limit of Equation (22) must be 0.

For $\delta = 1/2$, we start by establishing that

$$\lim_{q \to 1^-} \int_{-\infty}^{-\lambda} (z+\lambda)^2 f_Z(z)\,\mathrm{d}z = \lim_{q \to 1^-} \left(\sigma^2 \int_{-\infty}^{\frac{\theta}{\sigma}} y^2\,\phi(y)\,\mathrm{d}y + 2\theta\sigma \int_{-\infty}^{\frac{\theta}{\sigma}} y\,\phi(y)\,\mathrm{d}y + \theta^2 \int_{-\infty}^{\frac{\theta}{\sigma}} \phi(y)\,\mathrm{d}y\right) \tag{23}$$

is a positive constant since $\theta/\sigma \to -b$, $\sigma = \sigma_\varepsilon\sqrt{n}$, $\theta \to -\lambda$, and $\theta\sigma \to -\sigma_\varepsilon\sqrt{n}\lambda$. An identical argument can be made in the case of $\lim_{q \to 1^-} \int_\lambda^\infty (z-\lambda)^2 f_Z(z)\,\mathrm{d}z$. We then have

$$\lim_{q \to 1^-} \frac{1}{d^2} \int_{-\infty}^{-\lambda} (z+\lambda)^2 f_Z(z)\,\mathrm{d}z = \frac{C^+}{\lim_{q \to 1^-} d^2} = \frac{C^+}{0} = \infty,$$

where $C^+$ is some positive constant. And because $\lim_{q \to 1^-} \frac{1}{d} \operatorname{E} \hat{\beta}_j = \beta^*$ (Theorem 3.1), the limit of Equation (22) must be $\infty$.

Finally, for the case when $\delta > 1/2$, we have

$$
\begin{aligned}
\lim_{q \to 1^-} \frac{1}{d^2} &\left( \sigma^2 \int_{-\infty}^{\frac{\theta}{\sigma}} y^2 \, \phi(y) \, \mathrm{d}y + 2\theta\sigma \int_{-\infty}^{\frac{\theta}{\sigma}} y \, \phi(y) \, \mathrm{d}y + \theta^2 \int_{-\infty}^{\frac{\theta}{\sigma}} \phi(y) \, \mathrm{d}y \right) \\
&= \lim_{q \to 1^-} \left( \frac{n\sigma^2}{\left( n(q - q^2)^{1/2} + \lambda_2(q - q^2)^{2\delta - 1/2} \right)^2} \int_{-\infty}^{\frac{\theta}{\sigma}} y^2 \, \phi(y) \, \mathrm{d}y \right. \\
&\quad - \frac{2\sigma_\varepsilon \sqrt{n} \left( \beta^* n(q - q^2)^{1-\delta} - \lambda_1 \right)}{\left( n(q - q^2)^{3/4 - \delta/2} + \lambda_2(q - q^2)^{3\delta/2 - 1/4} \right)^2} \int_{-\infty}^{\frac{\theta}{\sigma}} y \, \phi(y) \, \mathrm{d}y \\
&\quad \left. + \left( \frac{-\beta^* n(q - q^2)^{1-\delta} - \lambda_1}{n(q - q^2)^{1-\delta} + \lambda_2(q - q^2)^\delta} \right)^2 \int_{-\infty}^{\frac{\theta}{\sigma}} \phi(y) \, \mathrm{d}y \right). \quad (24)
\end{aligned}
$$

Inspection of the exponents involving the factor $(q - q^2)$ shows that the first term inside the limit will dominate. And since the upper limit of the integrals, $\theta/\sigma \to 0$ as $q \to 1^-$, the limit must be $\infty$.

## D.2 Proof of Theorem B.2

We have

$$
\lim_{q \to 1^-} \operatorname{Var} \hat{\beta}_j = \lim_{q \to 1^-} \frac{\sigma^2}{d^2} \left( \frac{\sigma_\varepsilon \sqrt{n}(q - q^2)^{1/2 - \delta}}{n(q - q^2)^{1-\delta} + \lambda_2(q - q^2)^\delta} \right)^2 = \frac{\sigma_\varepsilon^2 n}{\lambda_2^2} \lim_{q \to 1^-} (q - q^2)^{1 - 4\delta},
$$

from which the result follows directly.

## D.3 Proof of Theorem 3.3

### D.3.1 Expected Value

Starting with the expected value, our proof follows a similar structure as in the proof for Theorem 3.1 (Section D). We start by noting the values of some of the important terms. As before we will drop the subscript $j$ everywhere to simplify notation. We have

$$
\mu = \beta^*(q - q^2)^\omega,
$$
$$
\sigma = \sigma_\varepsilon \sqrt{n(q - q^2)},
$$
$$
d = n(q - q^2) + \lambda_2(q - q^2)^\omega.
$$
$$
\theta = -\beta^* n(q - q^2) - \lambda_1(q - q^2)^\omega,
$$
$$
\gamma = \beta^* n(q - q^2) - \lambda_1(q - q^2)^\omega.
$$

$$
\frac{\theta}{\sigma} = -a\sqrt{q - q^2} - b(q - q^2)^{\omega - 1/2},
$$
$$
\frac{\gamma}{\sigma} = a\sqrt{q - q^2} - b(q - q^2)^{\omega - 1/2},
$$
$$
\frac{\theta}{d} = \frac{-\beta^* n - \lambda_1(q - q^2)^{\omega - 1}}{n + \lambda_2(q - q^2)^{\omega - 1}},
$$
$$
\frac{\gamma}{d} = \frac{\beta^* n - \lambda_1(q - q^2)^{\omega - 1}}{n + \lambda_2(q - q^2)^{\omega - 1}},
$$

First note the following limit (which is analogous to that in Equation (15)).

$$
\lim_{q \to 1^-} \frac{\theta}{\sigma} = \lim_{q \to 1^-} \frac{\gamma}{\sigma} = \begin{cases} -\infty & \text{if } 0 \le \omega < \frac{1}{2}, \\ -b & \text{if } \omega = \frac{1}{2}, \\ 0 & \text{if } \omega > \frac{1}{2}. \end{cases} \quad (25)
$$

As in Section D, we are looking to compute the following limit:

$$
\lim_{q \to 1^-} \operatorname{E} \hat{\beta} = \lim_{q \to 1^-} \frac{1}{d} \left( -\theta \, \Phi\left( \frac{\theta}{\sigma} \right) - \sigma \, \phi\left( \frac{\theta}{\sigma} \right) + \gamma \, \Phi\left( \frac{\gamma}{\sigma} \right) + \sigma \, \phi\left( \frac{\gamma}{\sigma} \right) \right). \quad (26)
$$

Starting with the terms involving $\Phi$ and assuming that the limit can be distributed, we have

$$
\begin{aligned}
\lim_{q \to 1^-} \left( -\frac{\theta}{d} \, \Phi\left(\frac{\theta}{\sigma}\right) + \frac{\gamma}{d} \, \Phi\left(\frac{\gamma}{\sigma}\right) \right) &= \lim_{q \to 1^-} \frac{\beta^* n + \lambda_1 (q - q^2)^{\omega - 1}}{n + \lambda_2 (q - q^2)^{\omega - 1}} \, \Phi\left(\frac{\theta}{\sigma}\right) \\
&\quad + \lim_{q \to 1^-} \frac{\beta^* n - \lambda_1 (q - q^2)^{\omega - 1}}{n + \lambda_2 (q - q^2)^{\omega - 1}} \, \Phi\left(\frac{\gamma}{\sigma}\right) \\
&= \begin{cases} 0 & \text{if } 0 \le \omega < 1, \\ \frac{\beta^* n}{n + \lambda_2} & \text{if } \omega = 1, \\ \beta^* & \text{if } \omega > 1. \end{cases}
\end{aligned}
\tag{27}
$$

The derivation of the first case in Equation (27) depends on $\omega$. For $0 \le \omega \le 1/2$, it stems from the facts that $\Phi(\theta/\sigma) \to 0$ and $\Phi(\theta/\sigma) \to 0$ as $q \to 1^-$ together with the existence of the $(q - q^2)^{\omega - 1}$ factor in both numerator and denominator. For $1/2 \le \omega < 1$, the terms cancel each other out. In the second case, when $\omega = 1$, the result stems from $\Phi(\theta/\sigma)$ and $\Phi(\gamma/\sigma)$ both tending to $1/2$ as $q \to 1^-$. And finally for $\omega > 1$, the terms involving the $(q - q^2)^{\omega - 1}$ factors vanish and again the values of the cumulative distribution functions tend to $1/2$.

Now, we turn to the terms involving the probability density function $\phi$. Again, we assume the limit is distributive so that

$$
\lim_{q \to 1^-} \frac{\sigma}{d} \left( \phi\left(\frac{\gamma}{\sigma}\right) - \left(\frac{\theta}{\sigma}\right) \right) = \lim_{q \to 1^-} \frac{\sigma}{d} \phi\left(\frac{\gamma}{\sigma}\right) - \lim_{q \to 1^-} \frac{\sigma}{d} \phi\left(\frac{\theta}{\sigma}\right).
\tag{28}
$$

Starting with the first term on the right-hand side of Equation (28), we have

$$
\lim_{q \to 1^-} \frac{\sigma}{d} \phi\left(\frac{\gamma}{\sigma}\right) = \frac{\sigma_\varepsilon \sqrt{n} \, \phi\left(\frac{\gamma}{\sigma}\right)}{n(q - q^2)^{1/2} + \lambda_2 (q - q^2)^{\omega - 1/2}}.
$$

For $0 \le \omega < 1/2$, this limit is 0 since the exponential terms in the numerator will dominate as $q \to 1^-$. For $\omega = 1/2$, we have the limit $\sigma_\varepsilon \sqrt{n} \, \phi(-b)/\lambda_2$. For $\omega > 1/2$, the limit is indeterminate of the type $0/0$. Let

$$
f_1(q) = \phi\left(\frac{\gamma}{\sigma}\right) \qquad \text{and} \qquad g(q) = n(q - q^2)^{1/2} + \lambda_2 (q - q^2)^{\omega - 1/2}
$$

and observe that $f_1$ and $g$ are differentiable and $g'(q) \ne 0$ for $q \in (1/2, 1)$. The derivatives are

$$
f_1'(q) = -\left( \frac{a}{2}(1 - 2q)(q - q^2)^{-1/2} - b(\omega - 1/2)(1 - 2q)(q - q^2)^{\omega - 3/2} \right) \frac{\gamma}{\sigma} \phi\left(\frac{\gamma}{\sigma}\right),
$$

$$
g'(q) = \frac{n}{2}(1 - 2q)(q - q^2)^{-1/2} + \lambda_2(\omega - 1/2)(1 - 2q)(q - q^2)^{\omega - 3/2}.
$$

Next, we find that

$$
\frac{f_1'(q)}{g'(q)} = \frac{-a + b(2\omega - 1)(q - q^2)^{\omega - 1}}{n + \lambda_2(2\omega - 1)(q - q^2)^{\omega - 1}} \left(\frac{\gamma}{\sigma}\right) \phi\left(\frac{\gamma}{\sigma}\right).
\tag{29}
$$

Taking the limit of Equation (29) and invoking L'Hôpital's rule yields

$$
\lim_{q \to 1^-} \frac{f_1'(q)}{g'(q)} = 0
$$

both when $1/2 < \omega < 1$ and $\omega \ge 1$ since $\gamma/\sigma$ tends to 0 as $q \to 1^-$ for $\omega > 1/2$ and the $\phi$ term tends to a constant, plus the fact that the remaining factor in the expression also tends to a constant since the terms involving $(q - q^2)^{\omega - 1}$ vanish when $\omega > 1$, are constant when $\omega = 1$, and cancel each other out in the limit when $\omega < 1$.

Finally, if we now consider the second term on the right-hand side of Equation (28), set $f_2(q) = \phi(\theta/\sigma)$, and perform the same steps as above, we find that the limits are the same in all cases, which means that the limits in Equation (28) cancel in the case when $\omega = 1/2$ and therefore that

$$
\lim_{q \to 1^-} \frac{\sigma}{d} \left( \phi\left(\frac{\gamma}{\sigma}\right) - \left(\frac{\theta}{\sigma}\right) \right) = 0
$$

for $0 \le \omega$. The limit in Equation (26) is given by Equation (27).

### D.3.2 Variance

The proof for the variance result is in many ways equivalent to that in the case of variance of the normalized unweighted elastic net (Section D.1) and we therefore omit many of the details here.

In the case of $0 \leq \omega < 1/2$, the proof is simplified since the $d^2$ term tends to $\infty$ whilst the numerator takes the same limit as in the normalized case, which means that the limit is 0 in this case. For $\omega = 1/2$, we consider Equation (23) and observe that it again tends to a positive constant whilst $\lim_{q \to 1^-} d^2 = 0$, which means that the limit of the expression, and hence variance of the estimator, tends to $\infty$. For $\omega > 1/2$, an identical argument for the expression in Equation (24) holds and the limit is therefore $\infty$ in this case as well.

### D.4 Proof of Theorem B.1

If $X_i \sim \mathrm{Normal}(\mu, \sigma)$, then $|X_i| \sim \mathrm{FoldedNormal}(\mu, \sigma)$. By the Fisher–Tippett–Gnedenko theorem, we know that $(\max_i |X_i| - b_n)/a_n$ converges in distribution to either the Gumbel, Fréchet, or Weibull distribution, given a proper choice of $a_n > 0$ and $b_n \in \mathbb{R}$. A sufficient condition for convergence to the Gumbel distribution for a absolutely continuous cumulative distribution function (Nagaraja & David, 2003, Theorem 10.5.2) is

$$\lim_{x \to \infty} \frac{d}{dx} \left( \frac{1 - F(x)}{f(x)} \right) = 0.$$

We have

$$\frac{1 - F_Y(x)}{f_Y(x)} = \frac{1 - \frac{1}{2}\,\mathrm{erf}\left(\frac{x-\mu}{\sqrt{2\sigma^2}}\right) - \frac{1}{2}\,\mathrm{erf}\left(\frac{x+\mu}{\sqrt{2\sigma^2}}\right)}{\frac{1}{\sqrt{2\pi\sigma^2}} e^{\frac{-(x-\mu)^2}{2\sigma^2}} + \frac{1}{\sqrt{2\pi\sigma^2}} e^{\frac{-(x+\mu)^2}{2\sigma^2}}}$$

$$= \frac{2 - \Phi\left(\frac{x-\mu}{\sigma}\right) - \Phi\left(\frac{x+\mu}{\sigma}\right)}{\frac{1}{\sigma}\left(\phi\left(\frac{x-\mu}{\sigma}\right) + \phi\left(\frac{x+\mu}{\sigma}\right)\right)}$$

$$\to \frac{\sigma(1 - \Phi(x))}{\phi(x)} \text{ as } n \to n,$$

where $\phi$ and $\Phi$ are the probability distribution and cumulative density functions of the standard normal distribution respectively. Next, we follow Nagaraja & David (2003, example 10.5.3) and observe that

$$\frac{d}{dx} \frac{\sigma(1 - \Phi(x))}{\phi(x)} = \frac{\sigma x(1 - \Phi(x))}{\phi(x)} - \sigma \to 0 \text{ as } x \to \infty$$

since

$$\frac{1 - \Phi(x)}{\phi(x)} \sim \frac{1}{x}.$$

In this case, we may take $b_n = F_Y^{-1}(1 - 1/n)$ and $a_n = \left(n f_Y(b_n)\right)^{-1}$.

## E Additional Experiments

In this section we present additional and extended results from the main section.

### E.1 Power and False Discoveries for Multiple Features

Here, we study how the power of correctly detecting $k = 10$ signals under $q_j$ linearly spaced in $[0.5, 0.99]$ (Figure 15(a)). We set $\beta_j^* = 2$ for each of the signals, use $n = 100\,000$, and let $\sigma_\varepsilon = 1$. The level of regularization is set to $\lambda_1 = n4^\delta/10$. As we can see, the power is directly related to $q_j$ and for unbalanced features stronger the higher the choice of $\delta$ is.

We also consider a version of the same setup, but with $p$ linearly spaced in $[20, 100]$ and compute normalized mean-squared error (NMSE) and false discovery rate (FDR) (Figure 15(b)). As before, we let $k = 10$ and consider three different levels of class imbalance. The remaining $p - k$ features have class balances spaced

evenly on a logarithmic scale from 0.5 to 0.99. Unsurprisingly, the increase in power gained from selecting $\delta = 1$ imposes increased false discovery rates. We also see that the mean-squared error depends on class balance. In line with our previous results, $\delta \in \{0, 1/2\}$ appears to work well for balanced features whilst $\delta = 1$ works better when there are large imbalances. In the case when $q_j = 0.99$, the model under scaling with $\delta = 0$ does not detect any of the true signals.

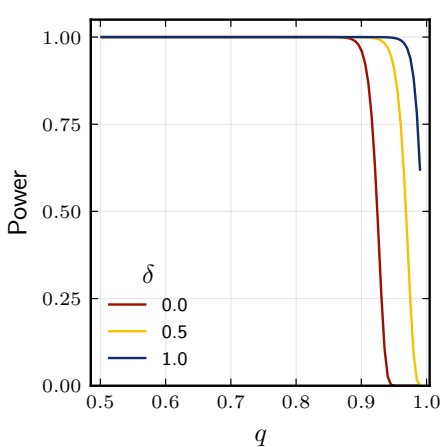

(a) The power (probability of detecting all true signals) of the lasso. In our orthogonal setting, power is constant over $p$, which is why we have omitted the parameter in the plot.

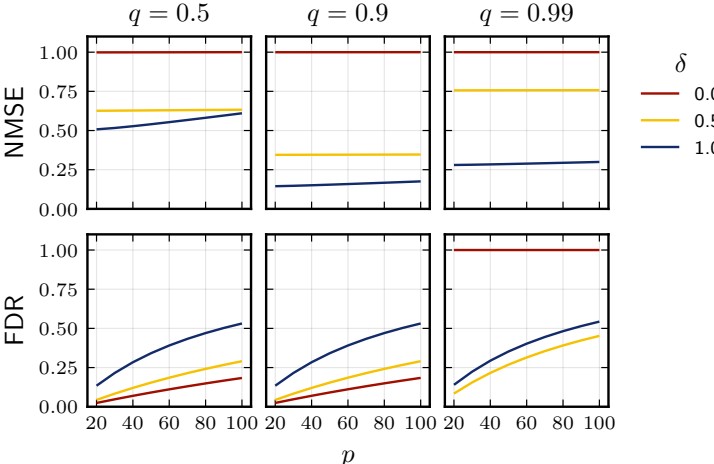

(b) NMSE and FDR: the rate of coefficients incorrectly set to non-zero (false discoveries) to the total number of estimated coefficients that are nonzero (discoveries).

Figure 15: Normalized mean-squared error (NMSE), false discovery rate (FDR), and power for a lasso problem with $k = 10$ true signals (nonzero $\beta_j^*$), varying $p$, and $q_j \in [0.5, 0.99]$. The noise level is set at $\sigma_\varepsilon = 1$ and $\lambda_1 = 0.02$.

### E.2 Support Size and Predictive Performance

In this section we analyze the support size of the lasso estimates for the experiment in Section 4.1.2. In Figure 16, we have, in addition to NMSE on the validation set, also plotted the size of the support of the lasso (cardinality of the set of features that have corresponding nonzero coefficients). Here we only show results for $\delta \in \{0, 1/2, 1\}$.

### E.3 Predictive Performance for Simulated Data

In this experiment, we consider predictive performance in terms of mean-squared error of the lasso and ridge regression given different levels of class balance ($q_j \in \{0.5, 0.9, 0.99\}$), signal-to-noise ratio, and normalization ($\delta$). All of the features are binary, but here we have used $n = 300$ and $p = 1000$. The $k = 10$ first features correspond to true signals with $\beta_j^* = 1$ and all have class balance $q$. To set signal-to-noise ratio levels, we rely on the same choice as in Hastie et al. (2020) and use a log-spaced sequence of values from 0.05 to 6. We use standard hold-out validation with equal splits for training, validation, and test sets. And we fit a full lasso path, parameterized by a log-spaced grid of 100 values[9], from $\lambda_{\max}$ (the value of $\lambda$ at which the first feature enters the model) to $10^{-2}\lambda_{\max}$ on the training set and pick $\lambda$ based on validation set error. Then we compute the hold-out test set error and aggregate the results across 100 iterations.

The results (Figure 17) show that the optimal normalization type in terms of prediction power depends on the class balance of the true signals. If the imbalance is severe, then we gain by using $\delta = 1/2$ or 1, which gives a chance of recovering the true signals. If everything is balanced, however, then we do better by not scaling. In general, $\delta = 1/2$ works well for these specific combinations of settings.

---

[9]This is a standard choice of grid, used for instance by Friedman et al. (2010)

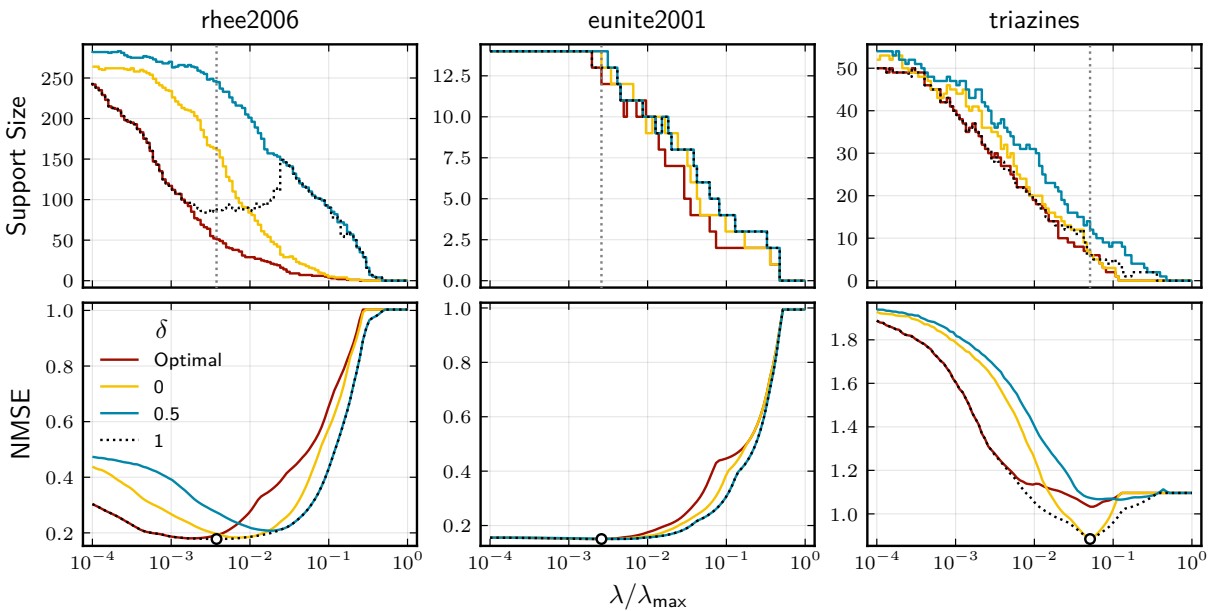

Figure 16: Support size and normalized mean-squared error (NMSE) for the validation set for the lasso fit to datasets rhee2006, eunite2001, and triazines across combinations of $\delta$ and $\lambda$. The optimal $\delta$ is marked with dashed black lines and the best combination of $\delta$ (among 0, 1/2, and 1) and $\lambda$ is shown as a dot.

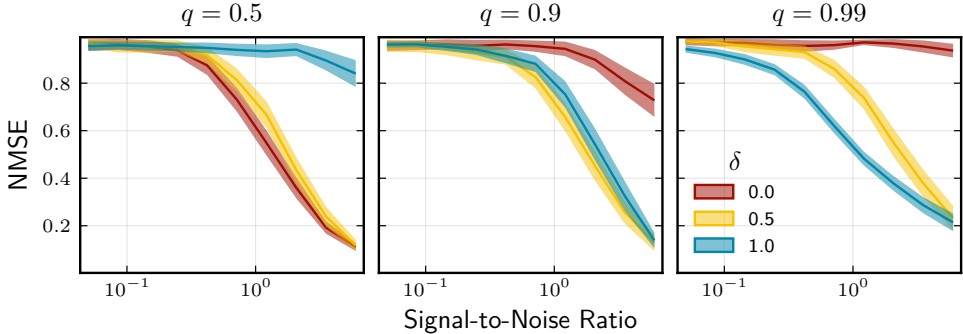

Figure 17: Normalized mean-squared prediction error in a lasso model for different types of normalization ($\delta$), types of class imbalances ($q_j$), and signal-to-noise ratios (0.05 to 6) in a dataset with $n = 300$ observations and $p = 1000$ features. The error is aggregated test-set error from hold-out validation with 100 observations in each of the training, validation, and test sets. The plot shows means and Student's $t$-based 95% confidence intervals.

### E.4 Comparisons of Normalization Methods on Real Data

In this section we present an extended and modified version of the experiment in Section 4.1.2, by considering several additional datasets, including datasets with binary responses to which we have fit $\ell_1$ and $\ell_2$-regularized logistic regression. Instead of only considering parameterization over $\delta$, we also extend the benchmarks to cover additional normalization types. For each of the datasets, we have fit the lasso and ridge for 10-times repeated 10-folds cross-validation over a grid of $\lambda$ and normalization method. In the case of the method we call "ours", we have extended the grid over $\delta$ as well, and return the results for the $\delta$ with lowest error.

The results are shown in Figure 18 and show that the optimal choice of normalization depends on the dataset and type of model. Using our strategy, which corresponds to standardization for continuous data and hyper-tuning over $\delta$ attains best results for most of the datasets, only ever slightly worse than the best performing method. Among the methods, $\ell_1$ normalization seems to perform poorly in general. Given the fact that that it corresponds to variance-scaling for binary data, which we have shown results in considerable variance, this is not particularly surprising.

### E.5 Dichotomization and Feature Selection

In this experiment, we attempt to study the effect of normalization choice in feature selection on datasets with dichotomized features and the following dichotomization scheme to convert the continuous features to binary.

Each variable was converted to a binary feature by comparing against threshold based on either historical standards, regulatory guidelines, or domain-specific knowledge of urban housing factors in the 1970s (Table 3).

Table 3: Dichotomization procedure and linear regression coefficients for the features of the Boston housing dataset used in the experiment in Section E.5.

| Feature | Dichotomization Rule | $\hat{\beta}_j^{\text{OLS}}$ | $q_j$ |
|---|---|---|---|
| Crime Rate | Above/below national U.S. average crime rate (1970–1971) | $-1.87$ | 0.90 |
| Zoning | Presence/absence of large lot zoning (residential lots over 25,000 sq.ft) | 1.96 | 0.26 |
| Business | Residential vs industrial areas (below/above 10% non-retail business acres) | $-2.41$ | 0.47 |
| Charles River | Original binary feature (borders river or not) | 5.12 | 0.07 |
| NOx Concentration | Above/below EPA air quality standard for NOx (53 parts per billion) | $-2.61$ | 0.52 |
| Rooms | Above/below typical family dwelling size (more/less than 6 rooms) | 3.49 | 0.66 |
| Age | Newer vs historic housing (less/more than 50% built before 1940) | $-1.10$ | 0.71 |
| Distance | Close vs far from employment centers (less/more than 5 miles) | $-4.76$ | 0.27 |
| Highway Access | Limited vs good highway access (accessibility index below/above 20) | 1.25 | 0.26 |
| Property Tax | Below/above Massachusetts average property tax rate (approximately \$12 per \$1000 in 1970s) | $-7.35$ | 0.97 |
| Pupil-Teacher Ratio | Below/above recommended educational value (16 students per teacher) | $-6.57$ | 0.83 |
| Demographics | More/less diverse population (above/below 85% white) | 3.25 | 0.95 |
| Lower Status | Middle-class vs lower-income areas (below/above 15% lower status population) | $-7.24$ | 0.32 |

Having dichotomized the data, we then first fit a standard linear regression model to the data, and use this as a proxy for feature importance. Then, for three types of normalization: $\ell_1$-normalization ($\delta = 1$),

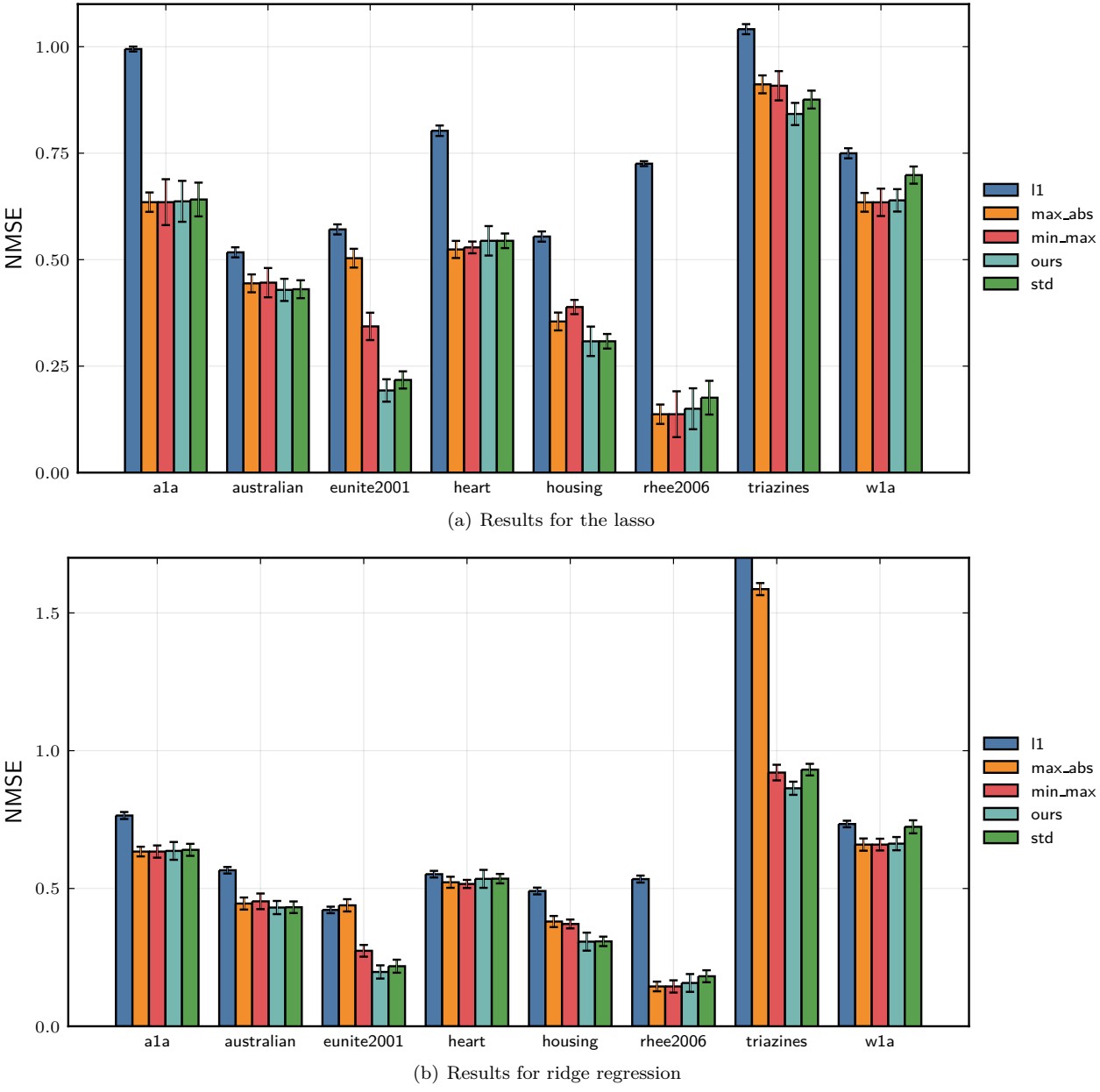

(a) Results for the lasso

(b) Results for ridge regression

Figure 18: Cross-validation error from 10-times repeated 10-folds cross validation for the lasso and ridge and various datasets and normalization strategies. The error is normalized mean-squared error (NMSE). In the case of datasets `a1a`, `w1a`, `heart`, `w1a`, and `australian`, we have fit regularized logistic regression and otherwise regularized linear regression. The error bars show 95% confidence intervals.

standardization, and min–max normalization, we fit a lasso model to the data and compute ranks of the features by checking at which point they enter the model.

Finally, we compare the ranks of the features in the lasso model to the ranks of the coefficients from the standard linear regression model, using the latter as the reference. We use four different metrics to compare the ranks: Spearman's rank correlation, Kendall's $\tau$, mean absolute difference, and normalized discounted cumulative gain (NDCG). The results are presented in Table 4. The results show that the normalization method using $\ell_1$-normalization ($\delta = 1$) best corresponds to the ranks of the linear regression coefficients.

Table 4: Comparison between ranks of ordinary least-squares coefficients and ranks given by the order of model entry along the lasso path for the Boston housing dataset. The metrics used are Spearman's and Kendall's rank correlations, normalized discounted cumulative gain (NDCG), and mean absolute difference (MAD). Best values are marked in blod face. For all measures except MAD, higher values are better.

| Metric | $\ell_1$-Normalization | Standardization | Min–Max/Max–Abs |
|---|---|---|---|
| Spearman | **0.7308** | 0.5714 | 0.5 |
| Kendall | **0.5128** | 0.4359 | 0.3846 |
| MAD | **2.0** | 2.7692 | 3.0769 |
| NDCG | **0.9515** | 0.9351 | 0.9186 |

## F  Categorical Features

This paper has focused on binary features, but since categorical features are typically encoded as a set of binary (dummy) features in regression settings, similar principles apply. For instance, categorical features will necessarily result in imbalanced representations with respect to the other features, given that one-hot encoding results in a set of mutually exclusive features. This means that the same issues with variance and regularization that we have discussed in the main part of the paper will also apply here.

We leave a thorough investigation of this to future work, but will here present a small experiment to illustrate another aspect of categorical features and class imbalance that is related to which reference category is chosen in one-hot encoding.

Consider a variable $Z \sim \mathrm{Categorical}(\boldsymbol{q})$, where $\boldsymbol{q} = (q_1, q_2, q_3)$. To represent this variable in a regression model, the standard approach is to one-hot encode it into a set of binary variables $X_1 = \mathbf{1}\{Z = 1\}$, $X_2 = \mathbf{1}\{Z = 2\}$, and $X_3 = \mathbf{1}\{Z = 3\}$, and then drop one of the variables to avoid redundancy. The dropped variable is called the reference category, and the coefficients of the remaining variables are interpreted in relation to this category.

Let us assume that we either use the first or the last category as reference, and that $q_1 = 1/20$, $q_2 = 1/2$, and $q_3 = 9/20$. In the first case, $\beta_{C:A}$ corresponds to the comparison between categories C and A. In the second case, $\beta_{A:C}$ corresponds to the comparison between categories A and C. In other words, these coefficients correspond to the same comparison, but the class balances of the features differ greatly. For a standard ordinary least-squares regression, this fact has no implication for the fit and either choice of reference category would yield the exact same fit. But for the lasso (or ridge), the choice of reference category *does* matter, since the regularization will affect the coefficients differently depending on the class balances of the features.

We set $\beta_{B:A} = \beta_{B:C} = 1/2$ and $\beta_{C:A} = \beta_{A:C} = 1$, and generate data according to the linear model $y = \boldsymbol{X}\boldsymbol{\beta}$. Then, we fit a lasso model to each case, using no normalization. The results are shown in Figure 19, where we plot the regularization paths for the two cases. The plot shows that $\beta_{C:A}$ enters the path earlier than $\beta_{A:C}$, even though they converge to the same value as $\lambda \to 0$. This means that the choice of reference category affects the order in which the features enter the model, and therefore potentially also affects feature selection.

## G  Summary of Data Sets

In Table 5 we summarize the datasets we use in our paper.

Table 5: Details of the real datasets used in the experiments. The median $q$ value refers to the median of the proportion of ones for the binary features in the data. Note that in the case of `housing`, there is only a single binary feature.

| Dataset | $n$ | $p$ | Response | Design | Median $q$ | Description |
|---|---|---|---|---|---|---|
| `eunite2001` | 336 | 16 | continuous | mixed | 0.856 | Mid-term load forecasting competition dataset (EUNITE 2001) using National Taiwan University's winning approach. Contains 15 features including 7-day historical loads (scaled) with winter-only training data from 1997-1998 to predict January 1999 daily maximum loads (Chen et al., 2004). |
| `housing` | 506 | 13 | continuous | mixed | 0.931 | Boston housing dataset with information about housing in the Boston area. Response is median value of owner-occupied homes in $1000s (Harrison & Rubinfeld, 1978). |
| `triazines` | 186 | 60 | continuous | mixed | 0.973 | Pharmaceutical dataset relating molecular structures of triazine derivatives to their ability to inhibit dihydrofolate reductase (Hirst et al., 1994; King et al., 1995). |
| `rhee2006` | 842 | 361 | continuous | binary | 0.995 | HIV-1 drug resistance data with protease and reverse transcriptase mutations as features. Response measures in vitro susceptibility to antiretroviral drugs (Rhee et al., 2006). |
| `leukemia` | 38 | 7129 | binary | continuous | | Gene expression data for leukemia patients. Classifies between acute myeloid leukemia (AML) and acute lymphoblastic leukemia (ALL) (Golub et al., 1999). |
| `australian` | 690 | 14 | binary | continuous | 0.557 | Credit approval dataset, originally from the StaLog database. The task is to predict credit approval using a number of different features (Quinlan, 1987; Henery & Taylor, 1992). |
| `heart` | 270 | 13 | binary | mixed | 0.678 | Heart disease dataset, originally from the StatLog database. The task is to predict the presence of heart disease from a number of features that have already been selected from a larger set of features (Henery & Taylor, 1992). |
| `a1a` | 1605 | 123 | binary | binary | 0.970 | Subset of Adult dataset derived from census data. Predicts whether income exceeds $50,000/year based on census information (Becker & Kohavi, 1996; Platt, 1998). |
| `w1a` | 2477 | 300 | binary | binary | 0.976 | Derived from web page data, classifying whether pages belong to specific categories (Platt, 1998). |

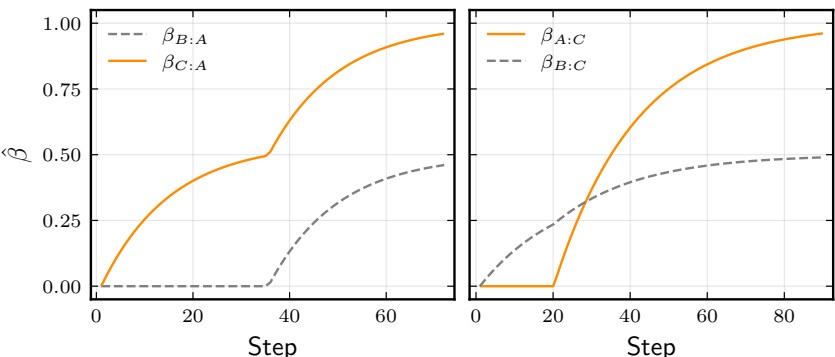

Figure 19: The coefficients along the regularization path for an example of fitting the lasso to a dataset with a categorical feature with three levels. The left plot shows the path when using category A as reference, and the right plot shows the path when using category C as reference. The coefficients $\beta_{C:A}$ and $\beta_{A:C}$ correspond to the same comparison, but enter the model at different points along the path.

We also visualize the distribution of class balance among all the binary features in Figure 20. We note that the class imbalance for many of these datasets is quite severe.

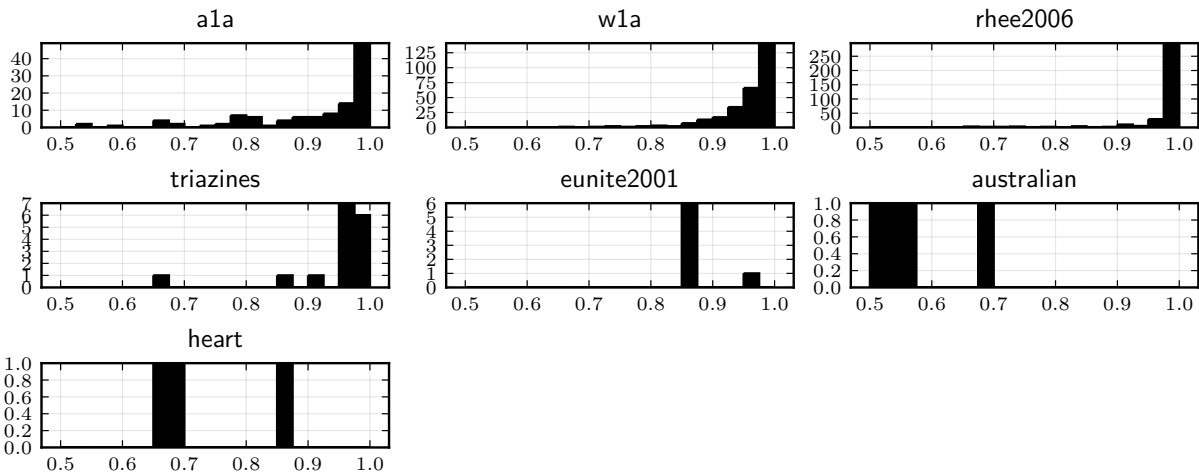

Figure 20: Histograms over the distribution of $q$ (class balance, that is, the proportion of ones) for the binary features in each of the datasets used in the paper.

