# OpenReview forum: "The Choice of Normalization Influences Shrinkage in Regularized Regression"
_TMLR — Accepted by TMLR_

### Review · Reviewer_2vk5 · 2025-07-22

**Summary Of Contributions:**

In this submission, the authors study the relationship between normalization schemes and fit coefficients for regularized linear regression (elastic net). In particular, they delve into the role that class imbalance plays in the effect of normalization. Their main findings include:

* Ridge regression and Lasso admit normalization schemes (feature std deviation and feature variance respectively) which remove class imbalance dependence from the final fit.
* In general elastic net does not admit a simple normalization scheme which does this.
* However, elastic net coupled with a specific coordinate-dependent regularization does have this property.
* In general, the choice of normalization scheme leads to a bias-variance tradeoff.

They conclude with experiments showing the tradeoffs between regularization strength and normalization scheme in the context of generalization error on real datasets.

**Audience:**

Yes

**Claims And Evidence:**

Yes

**Requested Changes:**

Light request: instead of labeling e.g. the design matrix X_{ij}, consider using mixed greek and roman indices like X_{\alpha j}. This improves the clarity of which quantities are in feature space vs data space.

The main request is to add a more detailed, separate section on practical lessons from the theory and the experiments, e.g. about how to choose the normalization scheme and/or sanity checks for making sure the normalization scheme is well selected. In particular it seems like choosing nontrivial $\delta$ was beneficial in the experiments, which may lead to recommendations for practitioners to use non-traditional normalization schemes.

**Strengths And Weaknesses:**

The paper is exceptionally well written, and the figures are all very well presented. I believe the findings are interesting for both linear regression but also machine learning in general.

The main weakness is the lack of clearly stated lessons for practitioners; some guidance is available but the main lessons appear to be scattered throughout the text. I believe consolidating these into a single discussion section could be valuable.

I believe the main contributions are original but I am not as familiar with the field; I will defer to other reviewers on this point.

---

> ### Author Response · Authors · 2025-10-14
>
> > Light request: instead of labeling e.g. the design matrix $X_{ij}$, consider
> > using mixed Greek and Roman indices like $X_{\alpha j}$. This improves the
> > clarity of which quantities are in feature space vs data space.
>
> Thank you for the suggestion, but we would like to argue that the current
> notation is standard and that mixing greek and roman indices might lead to more
> confusion, particularly since greek letters are often used for other purposes
> (e.g. parameters). We have, however, added a phrase to clarify the notation
> when it is first introduced. Please also note that we make scant use of the
> double-index notation in the paper, so we believe there is little potential for
> confusion.
>
> > The main request is to add a more detailed, separate section on practical
> > lessons from the theory and the experiments, e.g. about how to choose the
> > normalization scheme and/or sanity checks for making sure the normalization
> > scheme is well selected. In particular it seems like choosing nontrivial
> > $\delta$ was beneficial in the experiments, which may lead to recommendations
> > for practitioners to use non-traditional normalization schemes.
>
> We agree that this is a good idea and have added a new section (Section 5 in
> the revision) called "Practical Recommendations", which summarizes the main
> practical takeaways from our results.

---

### Review · Reviewer_NRCr · 2025-08-02

**Summary Of Contributions:**

This paper considers the effect of different input normalizations on the regression coefficients with Lasso, ridge or elastic regularization.

In more details, they assume the response $y$ is generated by
\begin{equation}
y = \beta^*\_0 + X \beta^* + \varepsilon
\end{equation}
with the target coefficients $\beta^*\_0, X \beta^*$ fixed, $X \in \mathbb{R}^{n\times p}$ the feature matrix, and $\varepsilon$ independent Gaussian noise.

The normalized feature matrix  $\tilde{X}$ is given by $\tilde{x}\_{ij}=(x_{ij}-c\_j)/s\_j$ for some centering coefficient $c_j$ and scaling coefficient $s_j$. This paper requires that the normalised features to be orthogonal, i.e. $X^\top X$ is diagonal.

Then the paper shows that the regression coefficient of binary features is biased depending on the normalization and the regularisation. This bias can be mitigated by corresponding normalisations. it also extends the discussions to gaussian features and mixed feature settings.

**Audience:**

Yes

**Broader Impact Concerns:**

No there is no or minor broader impact concerns I am aware of. This is a theoretical paper on linear regression.

**Claims And Evidence:**

Yes

**Requested Changes:**

In table 1, the $\hat{\beta}\_\text{max-ab}$ column is entirely zero. Is it correct or a mistake?

Also, could the authors explain the possibility or the difficulty how to extend the results to non-binary and non-Gaussian features?

**Strengths And Weaknesses:**

# Strength

This paper combines both empirical results and theoretical guarantees on feature normalisation in lasso, ridge and elastic regression, revealing the potential bias and possible mitigation with variance tradeoff. This paper also discussed the assumptions and limitations of the theoretical results in details.

# Weakness

A slight weakness is that the paper is based on strong assumptions in a very simple setting. Also, results only concerns the expected value of the regression coefficient at the limit of extremely unbalanced binary features.

---

> ### Author Response · Authors · 2025-10-14
>
> > A slight weakness is that the paper is based on strong assumptions in a very
> > simple setting.
>
> We agree that this is a limitation, but also want to stress that it is a first
> step in understanding the effect of normalization, and that ours is the only
> paper to have so far studied the issue in any capacity. And although the
> setting may seem simple, it is actually not trivial to extend the results to
> more complex settings.
>
> > Also, results only concerns the expected value of the
> > regression coefficient in the limit of extremely unbalanced binary features.
>
> This is not quite true. Equations (5), (6), and (8) are not
> asymptotic results. They hold for any $q$ and are
> closed-form expressions for expected value, variance, and probability
> of selection. In addition, please note that we have empirical
> results covering a wide range of simulated as well as real-world
> data.
>
> > In table 1, the $\hat{\beta}_\text{max-ab}$ column is entirely zero. Is it
> > correct or a mistake?
>
> Yes, that is in fact correct, and it underlines how strong the effect of
> normalization can be. We have clarified this in the part of the text
> where we discuss the table.
>
> > Also, could the authors explain the possibility or the difficulty how to extend
> > the results to non-binary and non-Gaussian features?
>
> We think it would be interesting to extend the results to other distributions
> too, but this would likely require extensive additional work, which we think is
> better suited for future work. We have, however, now included an additional
> experiment and short discussion on categorical features to the appendix
> (Appendix F).

---

### Review · Reviewer_hdbJ · 2025-09-28

**Summary Of Contributions:**

This paper studies how the choice of normalization affects shrinkage behavior in regularized regression methods, specifically lasso, ridge, and elastic net. While normalization (centering and scaling features) is standard practice to address feature scale sensitivity, this paper shows that the type of normalization applied has significant and often overlooked effects on model estimates, especially for binary features. They demonstrate theoretically and empirically that class balance in binary predictors directly influences coefficient estimates, with standard approaches such as standardization or max-abs scaling leading to systematic biases. For lasso and ridge, scaling binary features by their variance or standard deviation, respectively, can mitigate these biases, though at the cost of increased estimator variance. For the elastic net, no normalization fully removes this effect; instead, scaling penalty weights offers a solution. The study also extends to mixed binary–continuous data and interaction terms, showing that normalization implicitly governs relative penalization across feature types. Overall, the work highlights normalization as not merely a preprocessing step but a critical modeling choice that shapes bias–variance trade-offs and selection behavior in regularized regression.

- The paper shows that the class balance (proportion of ones) in binary features strongly affects coefficient estimates in lasso, ridge, and elastic net regression. It demonstrates that scaling by variance (lasso) or standard deviation (ridge) can reduce this bias, but only by trading off with increased estimator variance

- The authors prove that, unlike lasso and ridge, elastic net cannot eliminate class-balance bias through normalization alone. Instead, they propose scaling the penalty weights rather than the features themselves as an effective solution.

- The study extends its analysis to mixed binary–continuous designs and interaction terms, revealing that normalization choices implicitly control how regularization penalizes different feature types. They also show that commonly used normalization of interaction terms leads to biased estimates, and they provide an alternative scaling strategy that avoids this problem.

**Audience:**

Yes

**Claims And Evidence:**

Yes

**Requested Changes:**

- It would be better to include a discussion of highly sparse binary data (e.g., text or clickstream data), where normalization strategies may interact with sparsity constraints differently.

- The bias–variance trade-off can be quantified more clearly, e.g., error rates or feature selection accuracy in simulated settings with varying SNRs.

- It would be better to add more intuitive examples and visualizations (e.g., small toy datasets) to illustrate how normalization changes coefficient paths for non-technical readers.

**Strengths And Weaknesses:**

### Strengths

- The paper tackles an overlooked but important problem: how normalization choices affect shrinkage in regularized regression.
- It combines detailed theoretical analysis (deriving how class balance interacts with normalization) with extensive simulations and experiments on real datasets.
- The work provides insights for practices, for example, scaling binary features by variance or standard deviation depending on the method, or using weighted penalties for the elastic net.

### Weaknesses

- The study mainly focuses on binary and continuous features under least-squares loss. It would be better to discuss further how the results can be extended to categorical variables with more than two levels or broader classes of models.
- It would be better to discuss how the studies can be extended to some deep learning scenarios, for example, $\ell_2$ distance regularization for fine-tuning pretrained deep neural networks.

---

> ### Author Response · Authors · 2025-10-14
>
> > The study mainly focuses on binary and continuous features under
> > least-squares loss. It would be better to discuss further how the results can
> > be extended to categorical variables with more than two levels or broader
> > classes of models.
>
> We agree that it would be interesting to extend the results to a wider class of
> models and data types, but believe that this would require extensive additional
> work to be meaningful. We have, however, now included an additional experiment
> and short discussion on categorical features to the appendix (Appendix F).
> As we mention in the discussion of the paper, we also believe that
> an extension to generalized linear models is straightforward given
> that they depend directly on the same linear predictor.
>
> > It would be better to discuss how the studies can be extended to some deep
> > learning scenarios, for example, distance regularization for fine-tuning
> > pretrained deep neural networks.
>
> While we agree that this would be interesting and worthwhile, it would
> also require an altogether different analysis, which we believe is better
> suited for future work.
>
> > It would be better to include a discussion of highly sparse binary data
> > (e.g., text or clickstream data), where normalization strategies may interact
> > with sparsity constraints differently.
>
> We already cover highly sparse binary data. In fact, this is the primary target
> of our study (unbalanced binary features), so it is exactly the situation that
> both our theoretical and empirical work is focused on.
>
> > The bias–variance trade-off can be quantified more clearly, e.g., error rates
> > or feature selection accuracy in simulated settings with varying SNRs.
>
> Please note that we have closed-form expression for mean-squared error and
> feature selection probability through equations (5), (6), and (8). This is also
> visualized directly in Figures 3, 4, 13, and 14 (in the revised paper). We also
> study power and false-discovery rate empirically in the experiment in Appendix
> E.1 (see figure 14) and the effect of SNR in the experiment in Appendix E.3
> (see figure 17). Did you have any particular other experiment or result in
> mind?
>
> > It would be better to add more intuitive examples and visualizations (e.g.,
> > small toy datasets) to illustrate how normalization changes coefficient paths
> > for non-technical readers.
>
> Thank you for this suggestion! We have now included a new figure (Figure 2 in
> the revision), which shows coefficient paths along the regularization paths for
> balanced and imbalanced binary features under different types of normalization.

---

### Decision · Action_Editor_iWPy · 2025-11-11

**Recommendation:** Accept as is

**Additional Comments:**

The authors present this essential question in a simple and clear manner, I hope it can be exposed to more people and attract their interest on this largely ignored problem.

**Audience:**

Yes

**Audience Explanation:**

Generally, the theorists make idealized assumptions on the input. However, in practical scenarios, generally such assumptions do not hold, and we do some compromise to let the data roughly satisfy the assumptions. This paper indeed, pointed out that the general solutions of input normalization, may have different kinds of behaviors under different setup, which is a largely ignored issue for both theorists and practitioners. It is worth consideration for both of the world, and I hope this manuscript can be an initial point for this topic.

**Claims And Evidence:**

Yes

**Claims Explanation:**

This manuscript discussed an important but largely ignored problem in practice, namely how to normalize the input feature to obtain the best performance. It is indeed mathematically equivalent if there are no additional regularization presenting in the optimization objective, hence the authors focus on the case with explicit regularization. The authors focus on the case of imbalanced binary features, and discuss both in theory and in practice how the extent of the imbalance, together with the measurement noise, the optimization objective etc can affect the results. There are some minor concerns that the authors only focused on the simple setup i.e. linear model with imbalanced binary features. But as it reveals the largely ignored practical issue, I think this manuscript is indeed a good read.